# $\alpha$-PFN: Fast Entropy Search via In-Context Learning

**Herilalaina Rakotoarison** [* 1]    **Steven Adriaensen** [* 2]    **Tom Viering** [* 3]
**Carl Hvarfner** [4]    **Samuel Müller** [4]    **Frank Hutter** [5 6 2]    **Eytan Bakshy** [4]

## Abstract

Information-theoretic acquisition functions such as Entropy Search (ES) offer a principled exploration–exploitation framework for Bayesian optimization (BO). However, their practical implementation relies on complicated and slow approximations, i.e., a Monte Carlo estimation of the information gain. This complexity can introduce numerical errors and requires specialized, hand-crafted implementations. We propose a two-stage amortization strategy that learns to approximate entropy search-based acquisition functions using Prior-data Fitted Networks (PFNs) in a single forward pass. A first PFN is trained to be conditioned on information about the optima; second, the $\alpha$-PFN is trained to predict the expected information gain by training on information gains measured with the first PFN. The $\alpha$-PFN offers a flexible learned approximation, which replaces the complex heuristic approximations with a single forward pass per candidate, enabling rapid and extensible acquisition evaluation. Empirically, our approach is competitive with state-of-the-art entropy search implementations on synthetic and real-world benchmarks, while accelerating the different entropy search variants across all our experiments, with speed ups over 50x. Source code: https://github.com/automl/AlphaPFN.

## 1. Introduction

Bayesian Optimization (BO) is a method to maximize the output of a black-box function with as few as possible consecutive trials. It is especially beneficial when function evaluations are costly. BO finds use in various fields (Shahriari et al., 2015), such as optimizing the hyperparameters of large neural networks (Snoek et al., 2012; Feurer & Hutter, 2019). Here, the black-box function is the performance on a validation set given hyperparameter specifications, which necessitates a full training run for evaluation. It is thus of utmost importance to maximize performance in as few trials as possible. To achieve this, the canonical BO framework maintains a probabilistic regression surrogate model of the black-box function fit to the performance observations thus far, and maximizes an acquisition function quantifying the exploration-exploitation trade-off for the given posterior. Classical acquisition functions such as Expected Improvement (EI; Močkus, 1975) are inherently myopic, optimizing immediate improvements in observed values, and typically suboptimal in locating the optimum, e.g., in noisy or heterogeneous settings.

The information-theoretic acquisition functions (Villemonteix et al., 2009), such as Entropy Search (ES; Hennig & Schuler, 2012), offer a principled way to perform global optimization of Gaussian Process (GP) surrogate models (Williams & Rasmussen, 2006) by selecting queries that maximize the expected information gain regarding the location of the optimum. For example, ES naturally supports non-myopic, noise-robust, and cost-aware strategies that evaluate cheaper configurations not because they are expected to perform well themselves, but because they provide information about what performs best at higher evaluation costs (Klein et al., 2017). Although ES offers an elegant framework, its performance and runtime are hampered by complex handcrafted approximation schemes, as there is no simple analytical definition for GPs, unlike for classical acquisition functions such as EI.

Many efforts have been made to improve Entropy Search's computational efficiency and performance with GPs. Hernández-Lobato et al. (2014) show that it is possible to rewrite the Entropy Search acquisition function to depend on terms that are more tractable to estimate and approximate, resulting in an acquisition function which they call Predictive Entropy Search (PES). Wang & Jegelka (2017) proposed Max-value Entropy Search (MES) and most recently Hvarfner et al. (2022) and Tu et al. (2022) proposed

---

*Equal contribution    [1]University of Helsinki  [2]University of Freiburg  [3]Delft University of Technology  [4]Meta  [5]Prior Labs  [6]ELLIS Institute Tübingen. Correspondence to: Herilalaina Rakotoarison <rakoheri@ad.helsinki.fi>, Steven Adriaensen <adriaens@cs.uni-freiburg.de>, Tom Viering <t.j.viering@tudelft.nl>.

*Proceedings of the 43rd International Conference on Machine Learning*, Seoul, South Korea. PMLR 306, 2026. Copyright 2026 by the author(s).

Joint Entropy Search (JES). These methods adjust the original Entropy Search formulation to enable more efficient approximations and/or better BO performance. All of these improvements to ES still rely on handcrafted, sampling-based approximations, though. The runtime of BO is becoming an increasingly important issue, as practitioners also apply BO to black-box functions that are faster to query (so-called high-throughput optimization; Eriksson et al., 2019; Daulton et al., 2022; Maus et al., 2022).

Recently, it has been shown that Prior-Data Fitted Networks (PFNs; Müller et al., 2022), a transformer-based conditional neural process (Garnelo et al., 2018) can be used to speed up training and prediction of Gaussian Process regression models (Müller et al., 2023). The key idea is to use transformers to approximate posteriors for a prior: the transformer accepts the training data and test inputs, and outputs the posterior predictive distribution for each test sample. By training on millions of datasets sampled from a prior, the transformer meta-learns to perform Bayesian prediction in a single forward pass. Inference can be computationally much more efficient compared to fully-Bayesian GPs, which require the use of Markov Chain Monte Carlo (MCMC; Robert & Casella, 2005). Note that training PFNs has one-time upfront costs (typically on the order of a GPU day) – but these are amortized over millions of applications to different optimization problems, as inference is fast.

We explore how the same technique can be applied to approximate quantities used in Entropy Search. The Entropy Search acquisition functions (PES, MES and JES) all require many Monte Carlo samples and other manual approximations. Instead of deriving another approximation, our $\alpha$-PFN transformer *learns* to approximate the acquisition functions in a single forward pass. We evaluate the $\alpha$-PFN's performance against existing sampling-based approximations of PES, MES, and JES in terms of optimization quality (inference regret) and computational cost (optimization runtime). Our experiments cover two settings: Bayesian optimization with fully Bayesian GPs on synthetic benchmark functions and on real-world black-box hyperparameter optimization tasks from the LCBench (Zimmer et al., 2021) and HPO-B (Pineda-Arango et al., 2021) benchmark suites.

## 2. Background, Notation and Related Work

In this section, we introduce key concepts and related work.

### 2.1. Black-box Optimization and Bayesian Optimization

In Black-box Optimization (BBO), the goal is to maximize a function $f(x)$ over a domain $A$. We denote $x^* = \arg\max_{x \in A} f(x)$ and $f^* = f(x^*)$. We assume for simplicity that $A = [0,1]^d$. We only have black-box access to $f(x)$, i.e., we can only obtain function evaluations which

are typically corrupted with noise: $y = f(x) + \epsilon$. BBO is iterative: in each iteration $t$ a query $x_t$ is made to the black-box function and the corresponding observation $y_t$ is revealed. In Bayesian Optimization (BO), a Bayesian surrogate model, such as Gaussian Process or transformer, is fitted to the previously collected data, i.e., it is fitted on the training data $D_{trn} = \{(x_1, y_1), \ldots, (x_t, y_t)\}$ to predict the Posterior Predictive Distribution (PPD) $p(y_{tst}|D_{trn}, x_{tst})$ for a test point $x_{tst}$. BO uses an acquisition function $\alpha$ that operates on the predictive distribution and quantifies the desirability $\alpha(x, D_{trn})$ of evaluating some $x \in A$, and is used to decide which point to query next as $x_{t+1} = \arg\max_{x \in A} \alpha(x, D_{trn})$.

### 2.2. Entropy Search (ES)

The original Entropy Search (ES) method selects queries by maximizing the expected reduction in the entropy of the optimum location $x^*$ (Hennig & Schuler, 2012):

$$\alpha_{\text{ES}}(x, D_{\text{trn}}) = H(p(x^* \mid D_{\text{trn}}))$$
$$- \mathbb{E}_{y \sim p(y|D_{\text{trn}}, x)}[H(p(x^* \mid D_{\text{trn}} \cup \{(x,y)\}))] . \quad (1)$$

Here, $H(p(x^*|D_{trn}))$ represents the entropy of the posterior distribution over the optimum location given the observed data $D_{trn}$. The first term is constant with respect to query selection and can be ignored during optimization. The second term represents the expected entropy after obtaining the new observation $(x, y)$, where the expectation is taken over the predictive distribution $p(y|D_{trn}, x)$.

Computing $\alpha_{ES}$ exactly is generally intractable, as the entropy of the GP's maximizer does not have a closed-form expression and requires averaging over multiple samples of $y$. To approximate this, Hennig & Schuler (2012) propose two methods: Monte Carlo sampling and Expectation Propagation (Minka, 2001). However, both approaches are computationally expensive, making ES impractical for many real-world applications.

**Predictive Entropy Search (PES).** The entropy reduction in ES can be expressed in terms of mutual information (MI) between $x^*$ and $y$ conditioned on $D_{trn}$. Utilizing the symmetry of the MI in $x^*$ and $y$, Hernández-Lobato et al. (2014) propose to optimize

$$\alpha_{\text{PES}}(x, D_{\text{trn}}) = H(p(y \mid D_{\text{trn}}, x))$$
$$- \mathbb{E}_{x^* \sim p(x^*|D_{\text{trn}})}[H(p(y \mid D_{\text{trn}}, x, x^*))] . \quad (2)$$

which results in the same objective as ES (Equation 1) but allows more efficient approximation. The first term is analytically tractable since the posterior of the GP is Gaussian. Hernández-Lobato et al. (2014) propose a sequence of approximations to accurately estimate the entropy in the second term, and the outer expectation over $x^*$ is distributed according to $p(x^*|D_{trn})$. To obtain these, sample

paths from the GP posterior are approximated using random Fourier features (RFF, Rahimi & Recht, 2007). Each sample path is maximized to obtain draws of the posterior over $x^*$. The predictive posterior $p(y|D_{trn}, x^*)$ cannot be obtained exactly. Thus, conditioning on tractable alternatives, such as convexity at $x^*$, and constraints such as $f(x^*) \geq \max_{i \in [1,t]} y_i$, serves to approximate it.

**Max-value Entropy Search (MES).** Max-value Entropy Search (MES; Wang & Jegelka, 2017) aims to reduce uncertainty over the maximum function value, $f^*$, by selecting the query point that maximizes the expected information gain:

$$\begin{aligned}
\alpha_{\mathrm{MES}}(x, D_{\mathrm{trn}}) = {} & H(p(y \mid D_{\mathrm{trn}}, x)) \\
& - \mathbb{E}_{f^* \sim p(f^* | D_{\mathrm{trn}})}[H(p(y \mid D_{\mathrm{trn}}, x, f^*))].
\end{aligned} \quad (3)$$

Here, the expectation is taken over the posterior distribution of $f^*$ given $D_{trn}$. In addition to the RFF sampling approach for $f^*$ and $x^*$ proposed by Hernández-Lobato et al. (2014), Wang & Jegelka (2017) propose a simpler alternative for $f^*$ using a Gumbel distribution. Compared to PES, MES reduces the expectation from $d$ dimensions to one. Moreover, they assume that $p(y|D_{trn}, x, f^*)$ can be well-approximated by a truncated normal distribution, which enables an analytical entropy calculation. However, this assumption holds only in noiseless settings (Takeno et al., 2020; Nguyen et al., 2022). Due to the simpler approximation, MES is substantially faster than PES, and has seen multiple extensions, e.g. to parallel queries (Moss et al., 2021) (GIBBON-MES), noisy (Takeno et al., 2020; 2022) and multi-fidelity (Moss et al., 2021) problems.

**Joint Entropy Search (JES).** Either the distribution $p(x^*|D_{trn})$ or $p(f^*|D_{trn})$ might only provide a limited view of the posterior uncertainty over the optimum. Therefore Hvarfner et al. (2022) and Tu et al. (2022) propose to reduce the uncertainty on the joint distribution of the maximum value and its location:

$$\begin{aligned}
\alpha_{\mathrm{JES}}(x, D_{\mathrm{trn}}) = {} & H(p(y \mid D_{\mathrm{trn}}, x)) \\
& - \mathbb{E}_{(x^*, f^*) \sim p(x^*, f^* | D_{\mathrm{trn}})}[H(p(y \mid D_{\mathrm{trn}}, x, x^*, f^*))].
\end{aligned} \quad (4)$$

The expectation is approximated by sampling in the same manner as PES. For each sample, the pair $(x^*, f^*)$ is added to the GP's training set so that the posterior can be updated using regular GP machinery, conditioning either on a *noiseless* optimal value $f^*$ (Hvarfner et al., 2022), or a $y^* = f^* + \varepsilon$ containing observation noise (Tu et al., 2022). Both Hvarfner et al. (2022) and Tu et al. (2022) use a local constraint to condition on the maximum similar to MES. The resulting extended skew distribution (Nguyen et al., 2022; Hvarfner et al., 2022; 2023) does not admit a closed form for the entropy, and is therefore approximated either

by Monte Carlo sampling of the integral (Tu et al., 2022) or moment matching with a Gaussian (Moss et al., 2021; Hvarfner et al., 2022), lower bounding the MI (Moss et al., 2021).

**Fully Bayesian treatment for PES, MES and JES.** In practice, one may not know the ideal hyperparameters for the GP. A principled Bayesian would set hyperpriors on the hyperparameters, and integrate them out whenever possible. Because these integrals are expensive, typically approximations are used. Hernández-Lobato et al. (2014) were the first to give a fully Bayesian approximation to the acquisition function for ES (and Snoek et al. (2012) gave the first treatment in the BO literature). Hernández-Lobato et al. (2014) use slice sampling (Vanhatalo et al., 2013) to sample from the posterior over the GP hyperparameters, and this has also been applied to JES and PES (Hvarfner et al., 2022). The acquisition function is computed for each of these hyperparameter samples from the posterior and averaged. It would be more Bayesian to integrate out the hyperparameters first, and then compute a single acquisition function for the fully Bayesian model. However, such a fully Bayesian model would consist of a superposition of Gaussians, making this approach computationally hard. Our $\alpha$-PFN makes this approach computationally tractable.

### 2.3. Prior-data Fitted Networks (PFNs)

Prior-data Fitted Networks (PFNs; Müller et al., 2022) are transformer neural networks that learn to perform Bayesian predictions in a single forward pass by training on synthetic data sampled from a predefined prior $p(D)$. The prior defines a distribution over datasets $D$ of input and output pairs $(x, y)$. During training, datasets from the prior are split into two parts: a training and test set. The transformer takes the training set pairs $(x, y)$ as input (indicated as $D_{trn}$), and is trained to predict the correct outputs for the test set inputs. For simplicity of notation we always assume a test set of size one, and indicate it by $x_{tst}$ and $y_{tst}$. The transformer outputs a distribution for each test object, and is trained with cross-entropy. It can be shown that the transformer will approximate the posterior predictive distribution $p(y_{tst}|x_{tst}, D_{trn})$ for any prior $p(D)$ when trained on prior samples (Müller et al., 2022). At inference time, the training set and (unlabeled) test set are provided as input to the transformer, which then predicts test targets. Note that no gradient-descent takes place at inference time, but the PFN learns from new data in-context (during the forward pass).

The versatility of this paradigm has led to a variety of applications, such as forecasting time-series (Dooley et al., 2023) and learning curves (Adriaensen et al., 2023; Viering et al., 2024); and as a foundation model for tabular data (Hollmann et al., 2025). Most relevant to our work, PFNs have been used to accurately approximate Gaussian Process regression

for Bayesian Optimization (Müller et al., 2022; 2023).

In this work, we use the TabPFNv2 architecture (Hollmann et al., 2025), which improves upon the original PFN architecture used in PFNs4BO (Müller et al., 2023). Specifically, the original architecture requires zero-padding if features are not used, and can only be applied to dimensionalities seen during training. The v2 architecture processes each scalar cell of tabular data individually (a cell refers to a scalar value $x_i$ or $y$), encoding each to one embedding vector. Note that encoders for each $x_i$ are the same, but the encoder for $y$ is different. The layers of this architecture are composed of three modules: i) a self-attention across features and $y$ within each sample, ii) a self-attention between samples for each feature separately, and iii) a per-embedding MLP that processes each embedding separately.

### 2.4. Transformers for Black-box Optimization

Multiple previous works have proposed to use transformers for black-box optimization. Chen et al. (2022) proposed the use of transformers to predict points to be selected for Bayesian optimization, based on empirical traces of other Bayesian optimization algorithms utilizing handcrafted acquisition functions such as EI. Chen et al. (2017) also use GP sample paths and meta-learn acquisition functions, but do not focus on information-theoretic functions. Tiao et al. (2021) and later works by Song et al. (2022) propose the use of binary classification models to directly predict acquisition functions such as probability of improvement or expected improvement. Following a similar idea, end-to-end approaches (Volpp et al., 2020; Maraval et al., 2023) have also been proposed to simultaneously learn both the acquisition function and the surrogate model from scratch using reinforcement learning from trajectories. However, these latter two works focus on the transfer learning BO setting, where it is assumed that a family of related functions are available to learn from. In contrast, we do not use any transfer learning.

Chang et al. (2025) propose an architecture to perform inference conditional on latent variables, such as the maximum of a GP. They use a similar architecture as (Müller et al., 2022) to condition on $f^*$ and to predict the posterior over $f^*$, and then use Monte Carlo sampling at inference time to compute MES. In this work, we not only remove the slow Monte Carlo sampling at inference time by amortizing the acquisition computation itself, but also extend the approach to PES and JES. This is naturally harder for their approach, because they would need to predict $x^*$, which is higher-dimensional and requires autoregressive decoding due to strong dependencies.

Recent work has explored amortization of information-theoretic acquisition functions and related Bayesian active learning tasks. Most directly comparable, Igoe et al. (2026)

train equivariant neural networks to amortize Bayesian Experiment Design, of which Entropy Search is a special case. Hu et al. (2024) instead amortize the mutual-information computation itself; relative to that approach, the second stage amortization in $\alpha$-PFN offers further speed-ups since we do not need to approximate the expectation over optima at inference time. More broadly, Huang et al. (2026) jointly amortize Bayesian inference and active data acquisition using transformers, and Li et al. (2026) amortize safe active learning via neural policies pretrained on simulated nonparametric functions. All share our motivation of replacing expensive inference-time computation with a learned model, while differing in architecture, training prior, and the specific acquisition functions considered.

## 3. Entropy Search with PFNs

In the following we explain how we train $\alpha$-PFN, which predicts the acquisition values for PES, MES, and JES directly. See Figure 2 for an overview of the training setup. To generate training data for the $\alpha$-PFN, we first train an auxiliary (base) PFN. The base PFN is trained to make $y$ predictions (optionally) conditioned on information of the location and/or value of the optimum. Next to the ordinary PPD $p(y|D_{trn}, x)$, this base PFN can compute $p(y|D_{trn}, x, I)$ with $I = x^*$ for PES, $I = f^*$ for MES, and $I = (x^*, f^*)$ for JES, as illustrated in Figure 1. For each ES variant, we now train a second PFN model, the $\alpha$-PFN, which directly predicts the acquisition function. More specifically, $\alpha$-PFN is trained to approximate the full information gain distribution by using the base PFN's predictions, conditioned on each dataset's true optimum (precomputed, see below), as training targets, and at test time we recover the acquisition function (expected information gain), as the mean thereof.

**Pre-computing Gaussian Process prior data.** To construct our PFN models for GP inference we need to train the model on millions of dataset samples from a GP prior. Furthermore, we need to know $x^*$ and $f^*$ for each dataset, which is not feasible to compute for an exact GP sample. To make this feasible, we approximate the GP samples using Random Fourier Features (RFFs; Rahimi & Recht, 2007). The results of this precomputation are (approximate) samples, that can be queried efficiently for arbitrary $x$, on which we can employ gradient-based optimization to find approximate $x^*$ and $f^*$. See Appendix B for more details.

**Training the base PFN.** When we are training on our precomputed RFF GP data, we have access to $x^*$ and $f^*$, and feed these to the PFN to condition on them. We add one extra data point to the context of the PFN, which is encoded just like the other data points but using a different encoder, so that the PFN will learn to treat it differently. We randomly pass $x^*$ and $f^*$ with 50% probability such that a

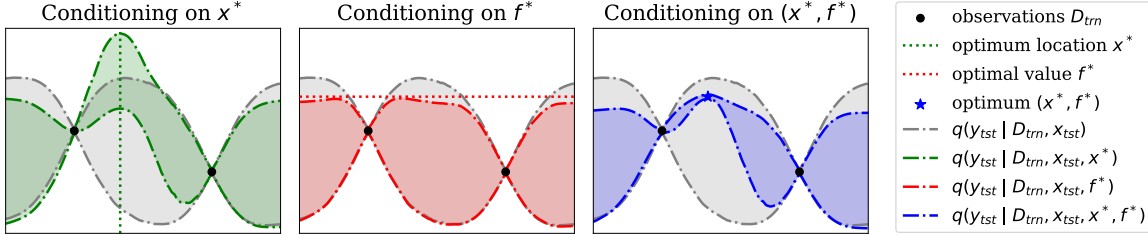

Figure 1. Our base PFN estimates the Posterior Predictive Distribution (PPD) conditioned on different types of information regarding the optimum: none (unconditional case), $x^*$, $f^*$, or both. Note that this example considers a specific optimum. In practice, the optimum, and therefore the information gain, is uncertain and the ES acquisitions compute *expected* gain. This is typically achieved by MC sampling, where $x^*$ and/or $f^*$ samples are obtained by optimizing GP posterior sample paths. We propose to learn to predict the gain distribution directly from $D_{trn}$ using another $\alpha$-PFN.

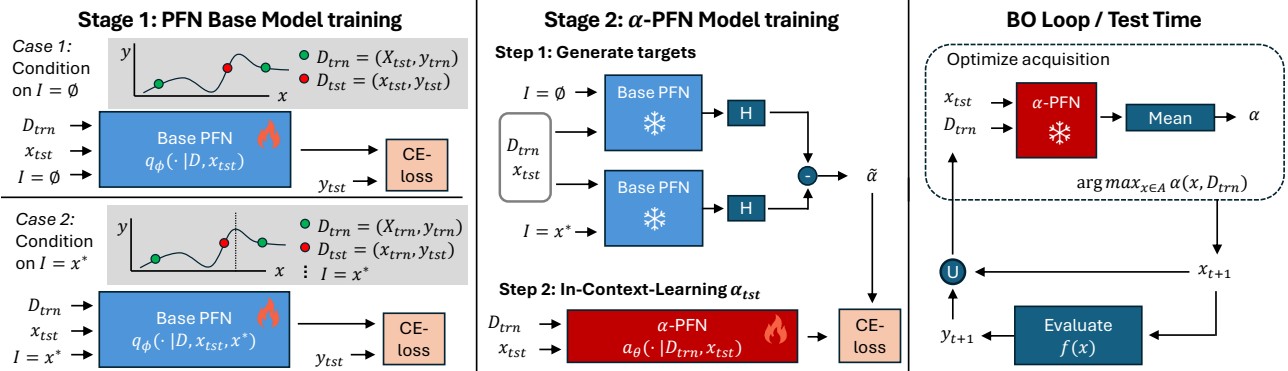

Figure 2. An overview of our pipeline. Left: we illustrate 2 out of 4 cases for the base PFN training. Middle: how to train the $\alpha$-PFN for PES. Right: how to use the $\alpha$-PFN at test-time in a BO loop. Note: PFNs were trained only once and reused in all our BO experiments.

single model is able to handle all four cases equally. This way we train our base PFN $q(y|x, D_{trn}, I)$, where $I$ can contain $x^*$ and/or $f^*$. For more details see Appendix C.2.

**Training $\alpha$-PFN.** After training the base model, we train a second PFN model, the $\alpha$-PFN, which takes the observation data $D$ and query point $x$ as input and predicts the acquisition $\alpha(x, D)$ directly. We train $\alpha$-PFN to use $D_{trn}$ and $x$ to predict

$$H(q(y|D_{trn}, x)) - H(q(y|D_{trn}, x, I)), \qquad (5)$$

where $I$ is extra information about the optimum. The (differential) entropy is computed analytically on the PFN's Riemann distribution (Müller et al., 2022).[1] During training, we can just feed in the $x^*$ and/or $f^*$ that were precomputed for our GP data. This info is only used to determine the prediction target during training and is *not required at test time for the $\alpha$-PFN*. $\alpha$-PFN will learn the distribution of this information gain, as this is a random variable, where the varying factor is the location and/or value of the optimum. The mean of this distribution that $\alpha$-PFN aims to learn coin-

cides with the PES/MES/JES acquisition in equations 2/3/4. We will formally prove this in the next section. This way, the $\alpha$-PFN avoids the need to sample $x^*$ and $f^*$ at test-time via MC samples. For more training details see Appendix C.

**Handling domain shift at optimization time.** Müller et al. (2023) train PFNs for BO by sampling inputs $x$ uniformly from the domain. However, during the process of BO, the query points $x$ tend to cluster around local optima. This domain shift in the pretraining procedure appears to adversely affect BO performance of PFNs in higher dimensions. To combat this, we train both base and $\alpha$-PFN with a sampling procedure that mimics the clustering behavior observed during the BO process. See Appendix E for more detail on how we implemented a fast heuristic to generate approximate BO-traces.

**Fully-Bayesian models.** To approximate ES for a fully-Bayesian GP, a GP with priors over its hyperparameters, we simply sample its hyperparameters before sampling from the GP at train time. This is the only minor change needed in our training pipeline and incurs virtually no additional overhead. When trained on fully-Bayesian data, our base model $q$ directly integrates out the uncertainty over the GP's hyperparameters. In contrast to the classical GP approach,

---

[1]Note that we predict the information gain of $y$, the noisy target, consistent with the original PES formulation (Hennig & Schuler, 2012; Hernández-Lobato et al., 2014).

we only compute one acquisition function, derived from the information gain of this fully Bayesian base model. This acquisition allows our ES-PFN to sample points to reduce the uncertainty with respect to the hyperparameters as well. As such, we can expect the fully-Bayesian $\alpha$-PFN to query points more efficiently in the fully-Bayesian case than the GP counterparts, since the GP ES variants ignore uncertainty over hyperparameters when computing the acquisition.

**Training procedure and resources.** We train one fully Bayesian base model and three $\alpha$-PFNs, one per ES variant, on 100M datasets sampled from the hyperprior with varying dimensionalities (1-6D) and varying lengthscales per dimensionality (Automatic Relevance Determination). Training the base model takes approximately 13 hours on $4\times$ NVIDIA H200, and each $\alpha$-PFN takes about 16 hours on $4\times$ NVIDIA L40S. For details regarding the training data generation, see Appendix B, and for model architecture and training, see Appendix C.

## 4. $\alpha$-PFN Approximates Entropy Search

We derive in what way $\alpha$-PFN approximates entropy search. To train $\alpha$-PFN, we use an additional PFN, our base model $q(y|x, D, I)$, where $D$ is the observation data, $x$ is the currently considered point and $I$ is extra information about the maximum of the function underlying $D$. We train the base model with a standard PFN objective, but additionally feed $I$ to it. The proof that $q$ approximates $p(y|x, D, I)$ is very similar to the PFN proof by Müller et al. (2022) and is given in Appendix A.

The actual $\alpha$-PFN $a_\theta(\cdot|D, x)$ takes $D$ and $x$ as input, and outputs a distribution, which has the different ES acquisition functions as its mean (depending on $I$). $\alpha$-PFN is trained to minimize

$$l_\theta = \mathbb{E}_{D,x,I}\left[-\log a_\theta(\tilde{\alpha}(D, x, I)|x, D)\right], \qquad (6)$$

where $\tilde{\alpha}(D, x, I) = H(q(y|D, x)) - H(q(y|D, x, I))$. To understand why this is the right loss, we re-examine the entropy search acquisition functions. See Section 2.2 for further reference on the definition of entropy search. Assuming $q$ approximates $p$ exactly, we can write PES, MES, JES as

$$\mathbb{E}_{\tilde{\alpha}\sim p(\tilde{\alpha}|D,x)}[\tilde{\alpha}] = \mathbb{E}_{I\sim p(I|D,x)}[\tilde{\alpha}(D, x, I)] \qquad (7)$$
$$= \mathbb{E}_{I\sim p(I|D,x)}[H(q(y|D, x)) \qquad (8)$$
$$- H(q(y|D, x, I))], \qquad (9)$$

where $I = x^*$ for PES, $I = f^*$ for MES and $I = (x^*, f^*)$ for JES. The change of variable in Equation 7 is justified by the law of the unconscious statistician, since we only use $\tilde{\alpha}$ inside an expectation and do not need its density explicitly.

**Proposition 4.1.** *The objective $l_\theta$ is equal to the KL divergence between $p(\tilde{\alpha}|D, x)$ and the $\alpha$-PFN's output up to a constant. We can see that the mean of $\alpha$-PFN's output distribution can then be used to obtain entropy search approximations, like in the acquisition function definition in Equation 7.*

*Proof.* This can be shown as follows

$$l_\theta = \mathbb{E}_{D,x,I}\left[-\log a_\theta(\tilde{\alpha}(D, x, I)|x, D)\right] \qquad (10)$$
$$= \mathbb{E}_{D,x,\tilde{\alpha}}\left[-\log a_\theta(\tilde{\alpha}|x, D)\right] \qquad (11)$$
$$= \mathbb{E}_{D,x}\left[\mathbb{E}_{\tilde{\alpha}}[-\log a_\theta(\tilde{\alpha}|x, D)]\right] \qquad (12)$$
$$= \mathbb{E}_{D,x}\left[\text{CE}(p(\tilde{\alpha}|D, x), a_\theta(\tilde{\alpha}|x, D))\right] \qquad (13)$$
$$= \mathbb{E}_{D,x}\left[\text{KL}(p(\tilde{\alpha}|D, x), a_\theta(\tilde{\alpha}|x, D))\right] + C, \qquad (14)$$

where CE is cross-entropy and KL is the Kullback-Leibler divergence. This shows that our training loss optimizes for $a_\theta$ to approximate $p(\tilde{\alpha}|D, x)$ under the assumption that $q$ is exact. Now we can use $a_\theta$ in Equation 7 to approximate all ES variants, as $\mathbb{E}_{\tilde{\alpha}\sim a_\theta(\cdot|x,D)}[\tilde{\alpha}] \approx \mathbb{E}_{\tilde{\alpha}\sim p(\tilde{\alpha}|D,x)}[\tilde{\alpha}]$. $\qquad\square$

## 5. Experimental Setup

The goal of the experiments is to show that $\alpha$-PFN is a practical and efficient alternative to GP-based approximations of Entropy Search. For this reason, we focus on a specific setting where PFN and GP use *the same exact* prior (defined in Appendix C.1), such that we can compare performances and runtimes of the two approaches directly. However, note that sharing the same prior also implies that PFN and GP should exhibit similar behavior and are not expected to show a large performance difference. Specifically, we do not expect GP-ES to be state-of-the-art on these benchmarks, so we do not claim $\alpha$-PFN variants to be either. Our evaluation stress tests $\alpha$-PFN. Most of the functions under evaluation will not match this restrictive prior, i.e., we evaluate $\alpha$-PFN out-of-distribution. We also test the extrapolation capabilities of $\alpha$-PFN to generalize on large dimensions (up to 16D) and context sizes (100 BO iterations), while being trained on smaller dimensions (up to 6D) and context sizes (up to 50).

**Baselines.** We compare $\alpha$-PFN against existing Entropy Search approximations in the BoTorch library (Balandat et al., 2020): JES (Hvarfner et al., 2022), MES-GIBBON (Moss et al., 2021) and PES (Hernández-Lobato et al., 2014). Since no fully Bayesian entropy search implementations exist, we compare against MCMC-ES instead, which approximates a posterior over the acquisition, rather than computing the acquisition of the fully Bayesian model. For our baselines we use NUTS (Hoffman & Gelman, 2014), which uses Hamiltonian Monte Carlo (HMC), again making use of the BoTorch library for JES, MES-GIBBON and PES. Furthermore, we include EI as a reference, to show the merits of using entropy-based acquisition functions.

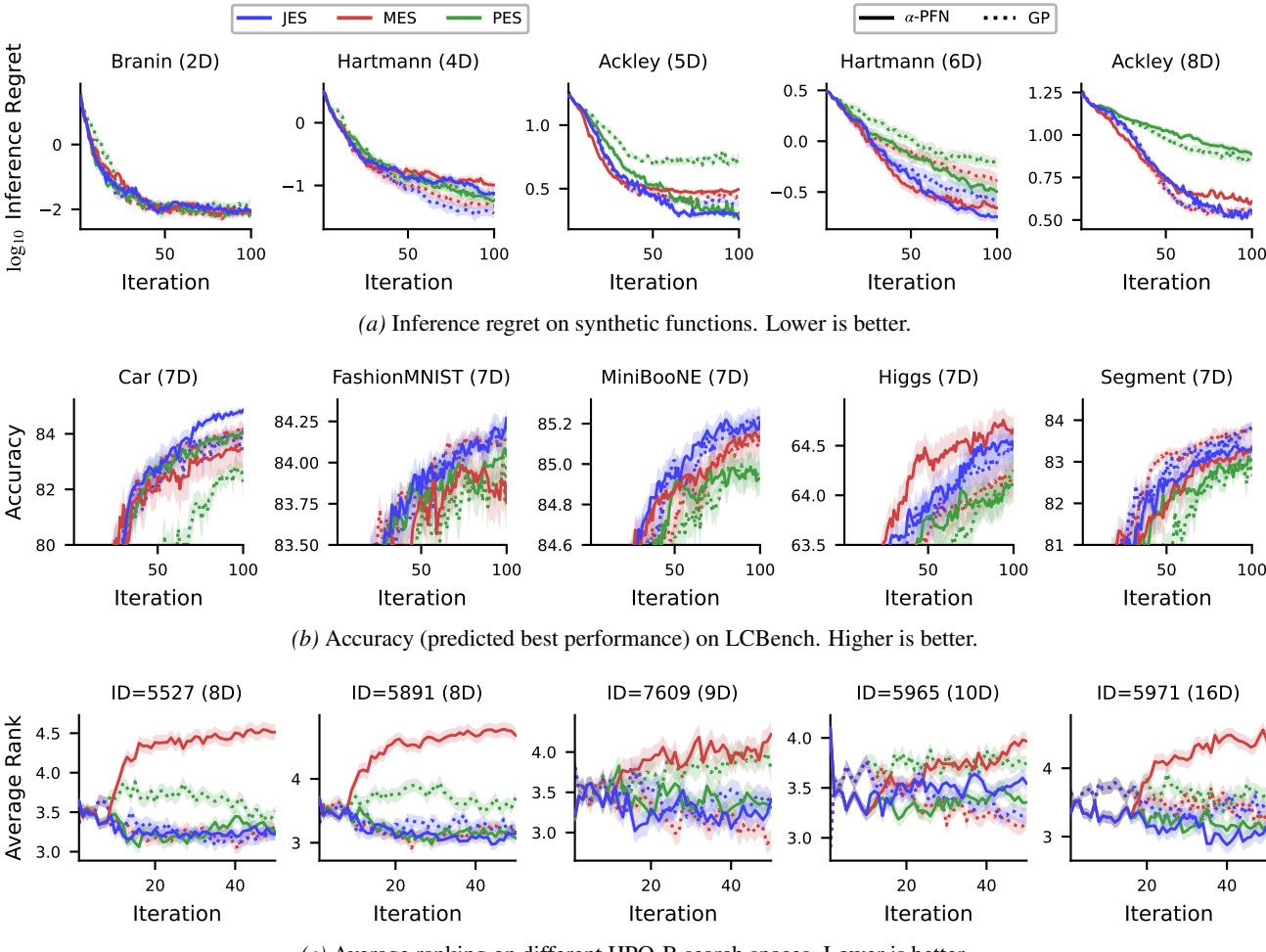

*(a)* Inference regret on synthetic functions. Lower is better.

*(b)* Accuracy (predicted best performance) on LCBench. Higher is better.

*(c)* Average ranking on different HPO-B search spaces. Lower is better.

*Figure 3.* Bayesian optimization performance comparison between GP-MCMC (NUTS) and $\alpha$-PFN across different synthetic test functions and real HPO benchmarks. The shaded area indicates one standard error.

**Evaluation datasets.** We evaluate on well-known black-box optimization benchmarks, including synthetic functions (Branin, Hartmann, Ackley) and real HPO tasks (LCBench (Zimmer et al., 2021), HPO-B (Pineda-Arango et al., 2021)). Test functions include both continuous and discrete domains. We incorporate one controlled out-of-distribution (OOD) study, where we add noise to the output that is significantly larger than noise during training. For further details (e.g., evaluation protocol) see Appendix D.

## 6. Results

For clarity, we include only regret-based comparison of JES, MES, and PES. Additional results, including comparison to EI (Figure 7), and pair-wise win-rate (Figure 8) are available in Appendix G. While EI, especially on HPO-B, yields strong performance, our goal is to compare ES methods. We show that EI PFN is competitive with GP EI (Appendix G).

**Synthetic Test Functions.** The results in Figure 3a are averaged over 30 repetitions. The PFN often closely matches the GP performance. This is supported by the moderate win-rates in Figure 8. For average regret, PES variants are generally competitive or superior. The JES variant underperforms on Hartmann 4D, as does the MES variant, which also shows degraded performance on large context sizes on Ackley 8D. On the other hand, all PFN variants perform better on Hartmann 6D, and JES additionally on Ackley 5D.

**Real-World HPO Tasks (LCBench, HPO-B).** In this experiment we consider optimizing real-world black-box optimization tasks, see Figures 3b-3c. Results are from 30 randomly initialized runs. Performances are often quite close, which is also reflected in the average rankings (over problems) that are quite high (last panel). On LCBench, $\alpha$-PFN variants often outperform the baselines, except on Segment. Overall, the JES-$\alpha$-PFN performs consistently better across all tasks. While MES-$\alpha$-PFN performs best in

*Table 1.* Runtime comparison (minutes) between GP-MCMC and $\alpha$-PFN across tasks. Speedup = GP/$\alpha$-PFN. Values represent the median cumulative runtime of model fitting and acquisition optimization across BO iterations. Domain: C = Continuous, D = Discrete.

| | Branin | Hartmann | Ackley | Hartmann | Car | Fashion | Higgs | MiniBooNE | Segment | Ackley | HPO-B-5527 | HPO-B-5891 | HPO-B-7609 | HPO-B-5965 | HPO-B-5971 |
|---|---|---|---|---|---|---|---|---|---|---|---|---|---|---|---|
| Dim | 2 | 4 | 5 | 6 | 7 | 7 | 7 | 7 | 7 | 8 | 8 | 8 | 9 | 10 | 16 |
| Domain | C | C | C | C | C | C | C | C | C | C | D | D | D | D | D |
| **GP Fully Bayesian** | | | | | | | | | | | | | | | |
| MES | 18.3 | 57.0 | 30.9 | 81.6 | 180.6 | 125.4 | 80.1 | 223.1 | 48.8 | 45.5 | 58.0 | 51.8 | 74.5 | 122.3 | 69.4 |
| JES | 26.8 | 55.0 | 60.4 | 172.3 | 259.7 | 123.4 | 102.7 | 150.0 | 66.8 | 78.9 | 75.9 | 82.6 | 41.5 | 128.1 | 86.2 |
| PES | 34.3 | 95.0 | 78.0 | 117.7 | 213.7 | 169.5 | 94.6 | 107.7 | 104.2 | 79.0 | 63.8 | 97.0 | 100.2 | 193.9 | 92.9 |
| **$\alpha$-PFN** | | | | | | | | | | | | | | | |
| MES | 5.4 | 11.1 | 12.6 | 18.9 | 21.0 | 16.7 | 28.1 | 28.4 | 30.0 | 23.4 | 10.5 | 1.7 | 1.1 | 10.1 | 2.9 |
| JES | 6.1 | 11.9 | 14.9 | 18.9 | 19.9 | 25.1 | 32.5 | 19.7 | 32.8 | 23.5 | 8.4 | 1.4 | 1.2 | 7.7 | 2.3 |
| PES | 5.3 | 9.3 | 15.5 | 20.3 | 20.9 | 15.0 | 19.6 | 26.1 | 25.4 | 20.6 | 11.7 | 1.7 | 1.4 | 9.4 | 2.9 |
| **$\alpha$-PFN Speedup ($\times$)** | | | | | | | | | | | | | | | |
| MES | **3.4** | **5.1** | **2.4** | **4.3** | **8.6** | **7.5** | **2.9** | **7.9** | **1.6** | **1.9** | **5.5** | **31.3** | **65.0** | **12.1** | **24.2** |
| JES | **4.4** | **4.6** | **4.0** | **9.1** | **13.1** | **4.9** | **3.2** | **7.6** | **2.0** | **3.4** | **9.1** | **58.7** | **34.4** | **16.7** | **37.6** |
| PES | **6.4** | **10.2** | **5.0** | **5.8** | **10.2** | **11.3** | **4.8** | **4.1** | **4.1** | **3.8** | **5.5** | **57.2** | **72.4** | **20.6** | **31.5** |

*Figure 4.* Noise ablation on Hartmann 4D and 6D, comparing the main setting ($\sigma_n = 0.316$) to a higher-noise OOD setting ($\sigma_n = 0.5$). $\alpha$-PFN degrades similarly to the corresponding GP baselines.

Higgs (LCBench), its performance is worse, often outperformed by the GP baseline, in particular on HPO-B.

**Timing Results.** Table 1 reports the median cumulative runtime (in minutes) for GP-MCMC and $\alpha$-PFN across all benchmarks. $\alpha$-PFN delivers substantial speedups across all tasks and acquisition functions, consistently outperforming GP-MCMC in computational efficiency. Speedups range from $1.6\times$ to over $72\times$, with HPO-B speedups commonly above $30\times$ and reaching beyond $70\times$. These results demonstrate that $\alpha$-PFN not only matches the optimization performance of GP-based entropy search methods but does so at a much lower computational cost, making $\alpha$-PFN practical for larger-scale applications.

**Out-of-distribution (OOD) Noise Ablation.** We evaluate the robustness of $\alpha$-PFN to OOD noise. We use a higher-noise setting which is unlikely under the training prior (Appendix C.1), $\sigma_n = 0.5$. Experiments are conducted on Hartmann 4D and 6D. As shown in Figure 4, performance degrades with larger noise for both GP and $\alpha$-PFN variants. Despite this out-of-distribution setting, we do not observe a clear additional failure mode for $\alpha$-PFN. Instead, it degrades at a similar rate as the GP baselines it approximates.

## 7. Summary, limitations, and future research

Our results demonstrate that PFNs can be used for Entropy Search. We show that $\alpha$-PFN is capable of simulating the state-of-the-art (JES) at reduced runtimes. The strong speed improvements highlight its ability to learn more efficient approximations than handcrafted alternatives. It is difficult to compare runtimes, as the hyperparameters of the baselines, such as the number of MC samples used, can influence runtimes significantly. We tried to set these reasonably.

The $\alpha$-PFN often matches the performance of the GPs, except for functions or datasets that are out of distribution. One way to mitigate this is through developing more diverse priors or transformations at test time. A major strength of our framework is its flexibility. While we used a GP-based hyperprior to align with standard ES methods, the PFN approach is agnostic to the choice of prior, e.g., Bayesian neural networks, ensembles, etc. could be used easily. Exploring alternative priors is a promising direction for future work. One bottleneck is that the $\alpha$-PFN needs to be retrained for each prior; something that may be resolved by methods like Whittle et al. (2026). So far, we have trained models up to 6 dimensions with training context sizes up to 50, and find that the PFNs can generalize to higher-dimensional problems or higher iteration counts. Yu et al. (2026) illustrate PFNs can scale to 500D. However, this is a significant investment which we postpone to future work.

In summary, our results position PFNs as a promising and general tool for acquisition function amortization. They invite further work on scaling, generalization to other Monte Carlo acquisition functions (Balandat et al., 2020), a more efficient implementation, and broader applications.

## Impact Statement

This work makes a fundamental contribution to the development of efficient acquisition functions for Bayesian optimization by replacing costly sampling-based approximations with learned approximations via Prior-data Fitted Networks (PFNs). As such, it does not directly interact with sensitive application areas or decision-making domains and is unlikely to pose immediate negative societal risks. By significantly reducing the computational overhead of Entropy Search, our approach lowers the resource requirements for effective Bayesian optimization. This has two positive implications: (1) it supports the democratization of advanced black-box optimization methods, enabling broader access to state-of-the-art tools without requiring large compute budgets; and (2) it contributes to reducing energy consumption in hyperparameter optimization and similar tasks, offering a modest but meaningful environmental benefit.

## Acknowledgements

Tom Viering, Steven Adriaensen, Herilalaina Rakotoarison, and Frank Hutter acknowledge funding by TAILOR, a project funded by EU Horizon 2020 research and innovation programme under GA No 952215. This work would not have happened without a research visit by Tom Viering made possible by the TAILOR Collaboration Exchange Fund. Frank Hutter acknowledges the financial support of the Hector Foundation. Steven Adriaensen, Herilalaina Rakotoarison, and Frank Hutter acknowledge funding by the state of Baden-Württemberg through bwHPC and the German Research Foundation (DFG), supporting project number 455622343 (bwForCluster NEMO 2); and by the European Union (via ERC Consolidator Grant 'Deep Learning 2.0', grant no. 101045765). This research was funded by the DFG under grant number 539134284, through EFRE (FEIH_2698644) and the state of Baden-Württemberg.

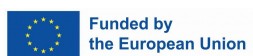 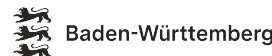

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

## A. Proof that the Base-PFN approximates the conditioned PPD

The loss used for training the conditional base-PFN is:

$$l_\theta = \mathbb{E}_{D \cup \{x,y\} \cup I} \left[ -\log q_\theta(y|x, D, I) \right]$$

Here, the $I$ is information that can be conditioned on. This is either: $x^*$, $f^*$ or both: $(x^*, f^*)$. Note that during the training, we sample $D$, $(x, y)$ and $I$ jointly. In practice, this is done by sampling the Gaussian Processes using the Random Fourier Features approximation, where we can control the identity of the Gaussian Process by a latent vector $w$. For such a Gaussian Process, we can bruteforce compute the maximum and its location, resulting in the information $I$ that we store together with $w$.

**Proposition A.1.** *The proposed objective $l_\theta$ is equal to the expectation of the cross-entropy between the true conditional PPD and its approximation $q_\theta$.*

*Proof.* The proof is analogues to the proof by Müller et al. (2022), where we in addition condition the true posterior on $I$ and the neural network $q_\theta$.

$$l_\theta = -\int_{D,x,y,I} p(x, y, D, I) \log q_\theta(y|x, D, I) = -\int_{D,x,I} p(x, D, I) \int_y p(y|x, D, I) \log q_\theta(y|x, D, I)$$

$$= \int_{D,x,I} p(x, D, I) \, H(p(y|x, D, I), q_\theta(y|x, D, I)) = \mathbb{E}_{(x,D,I) \sim p(x,D,I)} \left[ H(p(y|x, D, I), q_\theta(y|x, D, I)) \right]$$

where $H(p, q)$ indicates the cross entropy between $p$ and $q$. $\qquad\square$

## B. Pre-computing Approximate GP Data and their Maxima

### B.1. GP Approximation

The Random Fourier Feature (RFF) approximation obtains a sample path, which corresponds to a realization of a Gaussian Process. Such sample paths are represented by a weight vector $w$ in RFF space. To obtain a sample path, one samples $w$. The distribution of $w$ is determined by the kernel hyperparameters. Now, after fixing $w$, it is possible to evaluate the sample path at any position $x$. For a fixed $w$, it is thus possible to maximize it over the domain to estimate $x^*$ and $f^*$. Note that this is a difficult global optimization problem, which we solve approximately by an ensemble of optimizers that are restarted multiple times. Note that, the extension to the Fully Bayesian GP is natural. For the Fully Bayesian case we have a two-stage procedure: (1) we sample the kernel's hyperparameters, which determines the distribution over $w$. (2) we sample $w$ from this distribution.

For each setting, we generate millions of approximate samples from the corresponding GP prior (see Appendix C.1) using the Random Fourier Feature (RFF) approximation (Rahimi & Recht, 2007). We use 500 RFFs when computing the GP approximations.

### B.2. Identifying Approximate GP Maximizers

Computing the maximum of GP sample paths drawn via RFFs is a non-convex optimization problem. To identify the maximizers, we perform random search followed by first-order gradient-based optimization. First, we describe the optimizer and its hyperparameters, and afterward we describe the ensemble construction. We use either SGD or Adam and a batch of size `num_samples` points over the GP in parallel. The initialization is done either uniformly at random over the domain $[0, 1]^d$ (`resample_init` is False) or by trying a large number of points (`num_repeats` times `num_samples` points are tried), computing their function values, and keeping the best `num_samples` (if `resample_init` is True). Each optimizer is run for `n_iterations_max` iterations. During the optimization, we monitor the current best seen GP value so far and store it. If the current best point does not improve (compared with a tolerance `tol`), we increase a counter indicating the patience, and otherwise the patience counter is reset. After the patience counter reaches a value of `patience`, we decay the learning rate by a factor of `decay_factor`. If the learning rate is decayed more than `max_decays` times, we stop the optimization early. One should take care with points that move outside the domain during optimization, as optima are often located at the edge, especially for higher length scales / dimensions. If `clamp` is True, we always move the points back inside the domain by clamping. If `clamp` is False, we randomly initialize these points. If not specified, we use the default values for the optimizers as specified in PyTorch version 2.3.1.

*Table 2.* Hyperparameter grid values for building the GP Maximizer Ensemble.

| hyperparameter | grid values | hyperparameter | grid values |
|---|---|---|---|
| adam | [True, False] | max_decays | [1, 5, 10, 15] |
| init_lr | [0.001, 0.01, 0.1, 1] | tol | [1e-1, 1e-3, 1e-6] |
| resample_init | [True, False] | decay_factor | [0.1, 0.5, 0.99, 1] |
| num_samples | [50,200,1000] | clamp | [True, False] |
| n_iterations_max | [10, 100, 1000] | patience | [1, 5, 10, 20] |
| num_repeats | [10, 100, 1000] | | |

### B.3. Maximizer Ensemble Construction

We use the first 1000 GP sample paths to build the optimizer ensemble. We build a candidate set of 1000 optimizers with hyperparameters sampled from the grid as defined in Table 2. Because we want to determine the maxima for a large number of sample paths, we need to design a small ensemble with a low runtime yet good performance. We measure performance in terms of the regret of the ensemble, that is, the performance of the ensemble compared to the performance of the best optimizer in the candidate set. To reliably estimate the ensemble performance, we use 5-fold cross validation, where a train fold is used for the ensemble building, and the test fold is used for ensemble evaluation. The ensemble is constructed greedily by forward selection, where only candidates are considered that have an average runtime of less than 10 seconds per GP optimization. We keep adding ensemble members until the regret indicates we find the optimum with an approximate precision of 1e-6. We merge the ensembles together over the different training folds to come to a final ensemble.

## C. Training Details

We follow the same training procedure for training the base and $\alpha$-PFN models. As the training objective, we minimize the cross-entropy loss, using AdamW with a batch size of 100 datasets per GPU, a cosine decay learning rate schedule, and linear warmup over the first 1% of the training run. The GP sample used for a dataset is selected randomly from the pregenerated samples (see Appendix B), accounting for 90M GP samples for the fully Bayesian setting C.1, and is possibly reused multiple times. To counteract overfitting, and to encourage symmetry, we perform mirror and reflection augmentations on the domain $A$. Each dataset contains at most 150 points (context plus queries). The context size $C$ (train split) is fixed per batch and sampled uniformly from $[1, 49]$, and the context and query points themselves are drawn from the clustering-based trace generation procedure described in Appendix E.

We closely follow the architecture and training pipeline of previous PFN works (Müller et al., 2022; 2023), adopting the transformer architecture from Hollmann et al. (2025) that performs attention both between items and between features. The model is small (approximately 4.7M parameters) with 8 layers, each using an embedding size of 128, 4 attention heads, and 512 units in the hidden expansion layer. We use the regression head proposed by Müller et al. (2022) to model the output distribution. The output distribution is the Full-Bar Distribution (also called Riemann distribution) with Full-Support and 5000 bins.

### C.1. Training Prior

We train our models using a Fully Bayesian GP prior. The input dimension $d$ ranges from 1 to 6. We use the squared exponential kernel with an output scale $\sigma_f = 1$. The lengthscale per dimension is sampled independently (corresponding to Automatic Relevance Determination) from $P(D) = \text{LN}\left(\mu_0 + \frac{\log D}{2}, \sigma_0\right)$ with $\mu_0 = -0.75$ and $\sigma_0 = 0.75$ (this prior was inspired by Hvarfner et al. (2024)). We add zero-mean Gaussian noise with standard deviation $\sigma_n$ sampled from $\text{LN}(-4, 1)$. The mean $\mu$ of the GP is sampled from $\mathcal{N}(0, 0.5)$ per task.

### C.2. Base PFN Training Details

Recall that the base model is trained to condition on $x^*$, $f^*$ or both. During training, using pregenerated $x^*$ and $f^*$ values from GP sample paths (Appendix B), we randomly sample the conditioning to train the base model: (1) no conditioning, (2) condition on $x^*$, (3) condition on $f^*$, (4) condition on both. We train a single base model to support all four cases. These cases are all equally probable during training. Overall, we train our base model on 100M datasets (25M per GPU across 4

GPUs in data-parallel training).

## C.3. $\alpha$-PFN Training Details

The PFN model directly predicting the (expected) information gain closely follows the architecture and training of the base model. The main difference is that no conditioning tokens were used (and it does not take $I$ as input), and that the prediction targets for queries are not $y$, but the oracle information gain from equation 5. Note that we trained a $\alpha$-PFN for each ES variant. Each $\alpha$-PFN is trained on 100M datasets (25M per GPU across 4 GPUs), using similar data augmentation as the base model.

## C.4. Training Compute Resources

Pre-computing the GP sample paths and their maximizer (Appendix B) took approximately 40K CPU hours. Training the base PFN model (Appendix C.2) took approximately 13 hours of wall-clock time on $4\times$ NVIDIA H200 GPUs, for a total of about 52 H200 GPU-hours. Each of the three $\alpha$-PFN models (Appendix C.3) took approximately 16 hours of wall-clock time on $4\times$ NVIDIA L40S GPUs, i.e., about 64 L40S GPU-hours per model and 192 L40S GPU-hours across the three acquisition variants.

# D. Bayesian Optimization Experiment Details

The initial design consists of $d$ uniformly sampled points. At each BO iteration, the acquisition function is optimized using Botorch routine `optimize_acqf`, with the following hyperparameters: 128 uniform points as initial candidates and 10 restarts. We follow a similar optimization procedure to compute the maximizer of predictive posterior distribution, required for computing the inference regret.

The standard evaluation measure for methods using information-theoretic acquisition functions is the inference regret, defined as $f(x^*) - f(\hat{x}^*)$, where $\hat{x}^*$ is the maximizer of the posterior predictive distribution, i.e., $\hat{x}^* = \arg\max_{x \in A} \mathbb{E}[q(y|D, x)]$. This maximizer is approximated by performing gradient descent on $q(y|D, x)$, which can be either the surrogate GP or the PFN base-model, with a setup similar to that used for the acquisition function optimization.

We evaluate our method on three families of benchmarks:

- **Synthetic functions.** We use the standard BoTorch test suite (Balandat et al., 2020): Branin (2D), Hartmann (4D and 6D), and Ackley (5D and 8D). Each function is evaluated over 30 seeds with 100 BO iterations, and we report inference regret. We use $\sigma_n = 0.316$ to add noise to the synthetic functions.

- **LCBench surrogates.** We follow the LCBench setup of Hvarfner et al. (2025), which provides 7D surrogate tasks built on top of the LCBench (Zimmer et al., 2021) dataset. We use 5 tasks (`car`, `Fashion-MNIST`, `MiniBooNE`, `higgs`, `segment`), each evaluated over 30 seeds with 100 BO iterations, and we report the predicted best $f(\hat{x}^*)$ (computed similarly to the inference regret).

- **HPO-B.** We use the corrected version of HPO-B (Pineda-Arango et al., 2021) provided by FixedHPO-B (Müller, 2026)[2], which fixes the log-scale transformation across the original search spaces. We evaluate on 5 search spaces (5527, 5891, 7609, 5965, 5971) covering 8D to 16D problems, each evaluated over 5 seeds with 50 BO iterations, and we report the average rank of inference regrets across seeds.

For the fully Bayesian GP baselines, we use the fork of BoTorch maintained at `https://github.com/hvarfner/botorch/tree/pesfizx` (commit `b625f25c36ab306ec8de9e7d304f93f5be8916f5`), which provides fully Bayesian implementations of PES, MES, and JES. For the acquisition function optimization and inference optimization of $\alpha$-PFN models, we rely on the official BoTorch implementation (Balandat et al., 2020) (commit `3ebb38983732c68eeddb26f8f10c0a2e1501972d`).

---

[2] `https://github.com/SamuelGabriel/FixedHPO-B`

# E. Synthetic Optimization Trace Generation

## E.1. Motivation

Previous work (Müller et al., 2022; 2023; Rakotoarison et al., 2024) considered context and query points sampled uniformly. However, in real Bayesian Optimization (BO) traces, the context points follow a structured search pattern, dynamically balancing global and local search and often forming clusters around local optima. Likewise, uniform query points, in high dimensions, are unlikely to be near any of the context points, limiting the opportunity to learn to exploit the information they provide. Ideally, we would use actual optimization traces from the Entropy Search BO procedures. However, this would introduce a dependency on our surrogate model, leading to a chicken-and-egg problem. Alternatively, using BoTorch traces would restrict ourselves to the GP priors it supports. Instead, we propose a simple and efficient synthetic procedure that generates context and query points in a manner that mimics real BO traces.

## E.2. Procedure

Our synthetic optimization trace generation procedure, detailed in Algorithm 1, aims to replicate the characteristics of real BO traces by blending global and local search. The key components are:

- **Global search**: Points are sampled uniformly at random within the search space, with an additional probability $\epsilon$ of selecting a point exactly on the edge. This helps in exploring boundary effects which are important in higher dimensions as the optimizer increasingly often lies exactly on the edge.

- **Local search**: The next context points are drawn from a Gaussian distribution centered on the best observed context point so far. For query points, we select an arbitrary context point as the center. If the optimum $x^*$ is provided, it is sometimes chosen as the center, which facilitates learning the effect of conditioning on $x^*$.

- **Dynamic search adaptation**: BO dynamically transitions from global to local search over time. To model this, we define a local search probability $\alpha_i$ that linearly increases over $L$ steps. This ensures that earlier points explore the space globally, while later points refine the search locally.

- **Avoiding duplicate points**: Though not explicitly shown in the pseudocode, we ensure that no duplicate points occur in the trace. This frequently happens in corner regions. If a newly generated point coincides with an existing corner point, it is resampled.

This procedure effectively balances exploration and exploitation, producing synthetic traces that resemble real BO optimization trajectories while remaining computationally efficient.

## E.3. Trace Generation Ablation

We ablate the trace generation procedure by training $\alpha$-PFN either with the clustered traces from Algorithm 1 or with uniformly sampled context and query points. The results in Figure 5 show that clustered traces become more important as the dimensionality increases. This result suggests that a careful design of the training distribution (of context and query points) is required to avoid complete domain shift at inference time (during Bayesian optimization).

---

**Algorithm 1** Generate Optimization Trace

---

1: **Inputs:**
2:     $L$: length of the trace
3:     $C$: number of context points (i.e., we have $L - C$ query points)
4:     $d$: dimension of search space
5:     $GP\_sample$: function to evaluate GP values
6: **Outputs:**
7:     $trace$: matrix of shape $(L, d)$ with context/query points in the trace
8:     $y$: vector of length $C$ with function values for context points
9: **Procedure:**
10: Initialize $trace \leftarrow$ zero matrix of size $(L, d)$
11: Initialize $y \leftarrow$ zero vector of size $C$
12: $\epsilon = (1 - u^{\frac{d}{6}})$ with $u \sim U(0, 1)$ {Sample edge probability}
13: $\sigma \sim \text{LogNormal}(-3, 0.5)$ {Sample local search step size}
14: Sample initial / final local search probability:
15:     $\alpha_0 = \min(v_1, v_2, v_3)$
16:     $\alpha_L = \max(v_1, v_2, v_3)$
17:     with $v_1, v_2, v_3 \sim U(0, 1)$
18: Start trace from a random point in search space:
19:     $best\_point = trace[0] \leftarrow clip(w, 0, 1)$ with $w \sim U^d(-\frac{\epsilon}{2}, 1 + \frac{\epsilon}{2})$
20:     $y_{best} = y[0] \leftarrow GP\_sample(trace[0])$
21: **for** $i = 1$ to $L - 1$ **do**
22:     $\alpha_i \leftarrow \alpha_0 + (\alpha_L - \alpha_0) \cdot (i/L)$ {Determine local search probability}
23:     $local \leftarrow \text{Bernoulli}(\alpha_i)$ {Determine local or global search}
24:     **if** $local$ **then**
25:         **if** $i < C$ **then** $inc \leftarrow best\_point$ {Sample near the best point thus far}
26:         **else** Choose $inc$ randomly from context points ($trace[: C]$)
27:         $trace[i] \leftarrow clip(x, 0, 1)$ with $x \sim \mathcal{N}^d(inc, \sigma^2)$
28:     **else**
29:         $trace[i] \leftarrow clip(x, 0, 1)$ with $x \sim U^d(-\frac{\epsilon}{2}, 1 + \frac{\epsilon}{2})$ {Global search}
30:     **end if**
31:     {For context point, sample value and update best}
32:     **if** $i < C$ **then** $y[i] \leftarrow GP\_sample(trace[i])$
33:         **if** $y[i] > y_{best}$ **then** $best\_point \leftarrow trace[i]; y_{best} \leftarrow y[i]$
34: **end for**
35: **return** $trace, y$

---

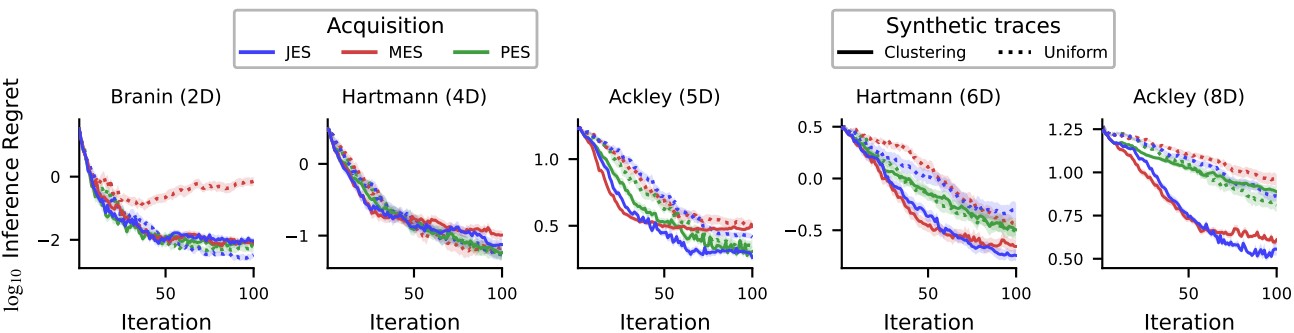

*Figure 5.* Trace generation ablation comparing clustered traces from Algorithm 1 with uniformly sampled context and query points.

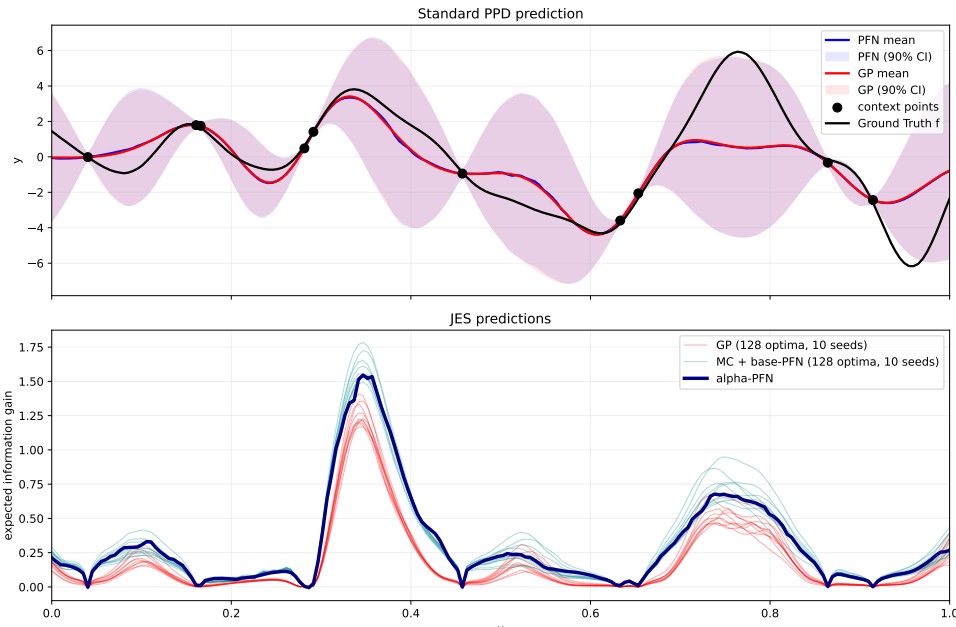

*Figure 6.* Qualitative comparison of GP, Monte Carlo base-PFN, and $\alpha$-PFN estimates for a 1D setting. For this setting, we use a non-fully Bayesian GP (and we trained a base-PFN and $\alpha$-PFN specifically for this ablation). This makes the acquisition function of the GP more comparable to the PFN. Top: GP and base-PFN posterior predictive distributions. Bottom: JES estimates, with Monte Carlo curves shown for 10 random seeds.

## F. Qualitative Information Gain Comparison

We compare qualitatively the information gain estimates between GP and PFN. For that, we consider JES for a GP prior with fixed hyperparameters: mean 0, lengthscale 0.05, noise variance 0.01, and output variance 10. This setting deviates significantly from the setting in the main body, as here we do not consider the Fully Bayesian setting. It means new PFN models are trained on this GP prior specifically (1D only). We consider fixed hyperparameters, since hyperpriors are treated very differently in the GP / PFN approaches, making the resulting acquisitions incomparable. Figure 6 shows a comparison of GP, Monte Carlo base-PFN, and $\alpha$-PFN. Note that Monte Carlo base-PFN uses the base-PFN for conditioning, but averages across (128) MC samples from the posterior over $(x^*, f^*)$, the exact process that $\alpha$-PFN amortizes. We observe that while acquisition peaks are roughly aligned, the conditional base-PFN provides "richer" information gain estimates. It also shows that MC-based estimates (for 10 different seeds, each based on 128 different optima samples) are highly variable in regions of interest, noise that $\alpha$-PFN is pretrained to model and average out.

## G. Additional Results

### G.1. Detailed Results

Figure 7 provides detailed results across all benchmarks, including PES, MES, JES and EI.

### G.2. Win Rate Comparisons

Figure 8 provides win rate comparisons. The win rate measures the fraction of tasks where $\alpha$-PFN achieves the better performance compared to the GP fully Bayesian models for each acquisition function variant, offering a complementary view to the average regret and ranking metrics. Except on MES-$\alpha$-PFN on HPO-B, $\alpha$-PFN achieves competitive performance to the GP fully Bayesian models, which is consistent with the fact that they share the same prior.

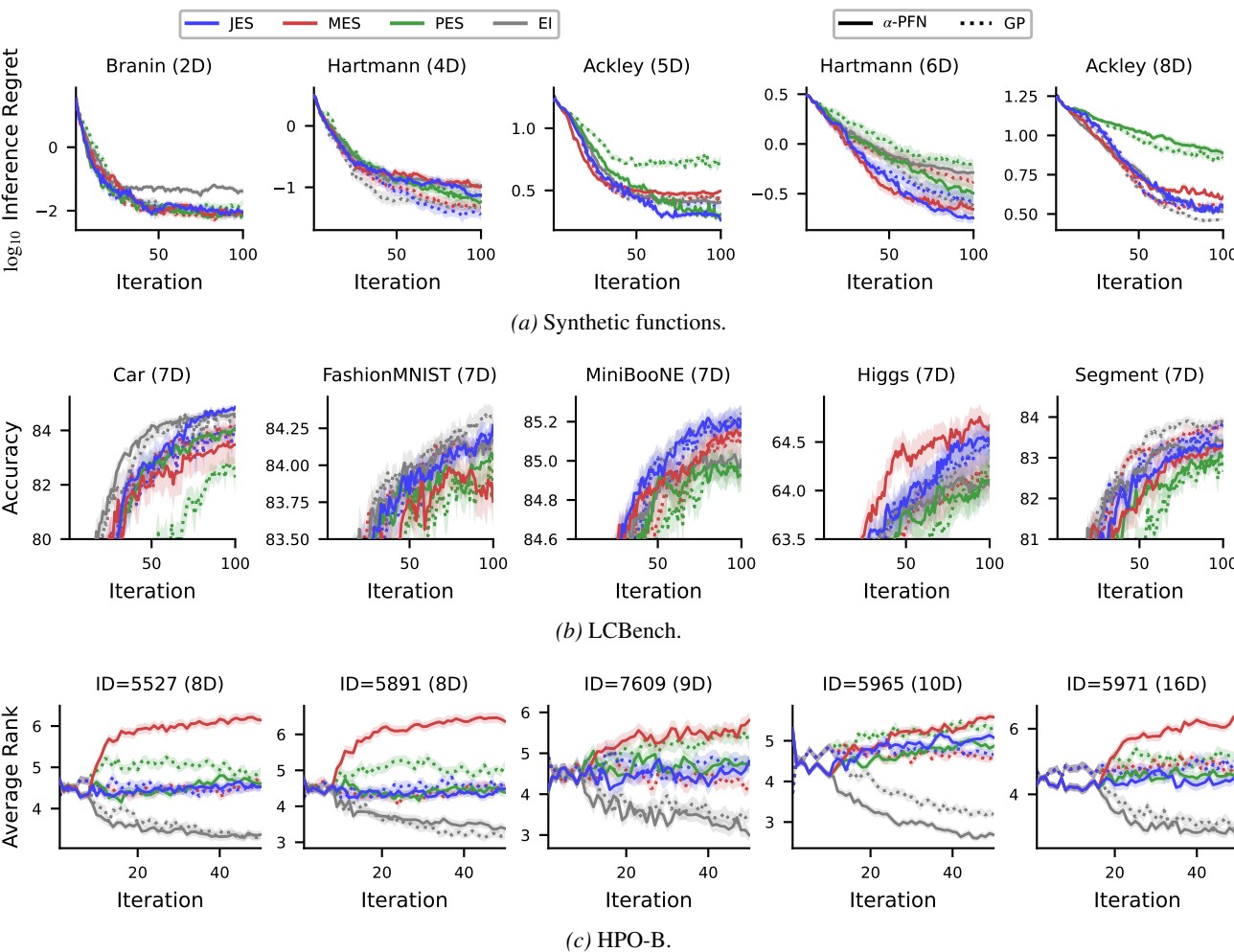

*(a)* Synthetic functions.

*(b)* LCBench.

*(c)* HPO-B.

*Figure 7.* Full results across all benchmarks.

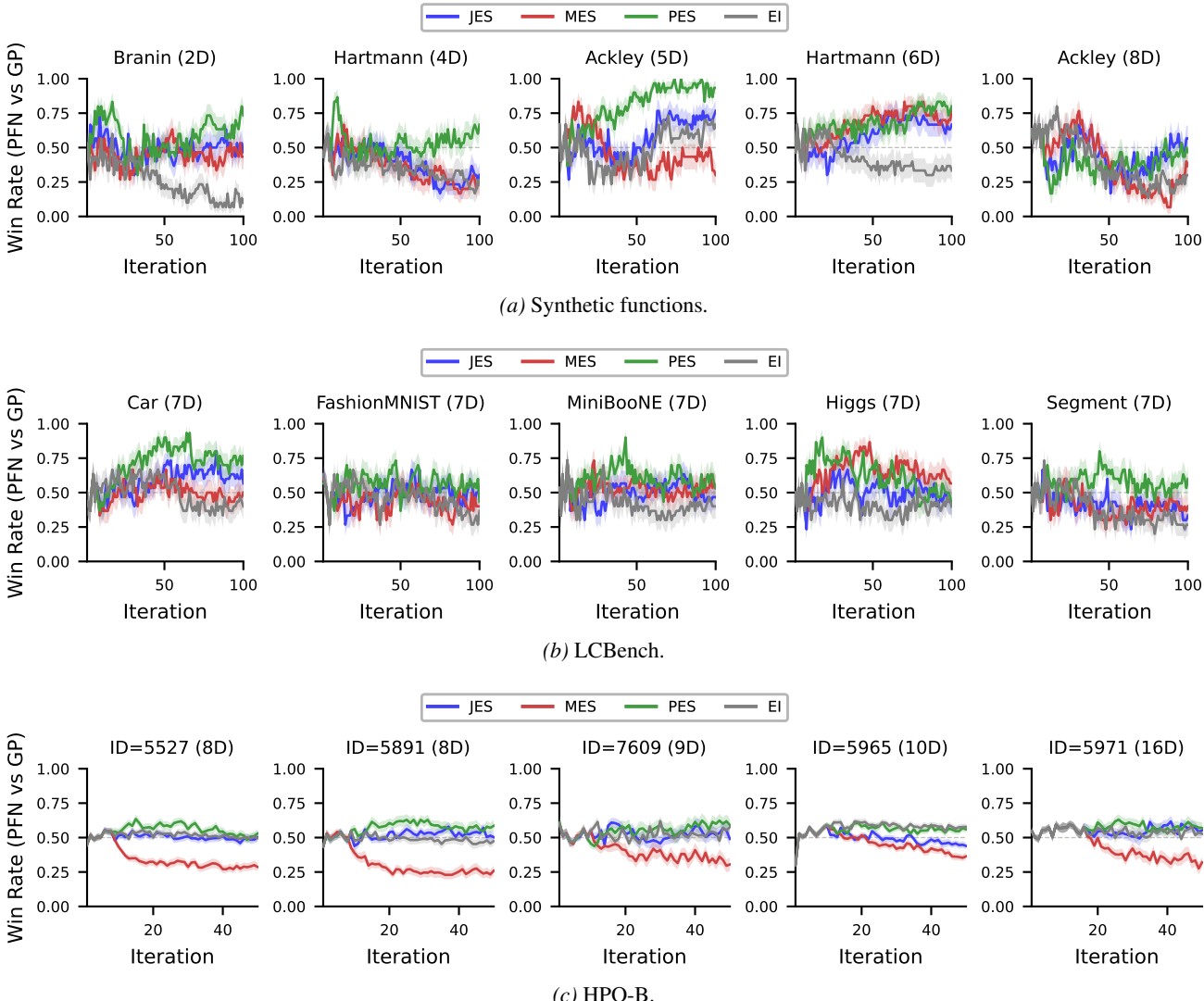

*Figure 8.* Win rate comparison across different benchmarks.

# H. Contribution Statement

We follow the CRediT taxonomy (Allen et al., 2014) for describing author contributions [3]

- **Herilalaina Rakotoarison**: Investigation, Methodology, Software, Writing – review & editing, Validation, Visualization.

- **Steven Adriaensen**: Conceptualization, Investigation, Methodology, Writing – original draft, Writing – review & editing, Visualization.

- **Tom Viering**: Conceptualization, Data curation, Visualization, Writing – original draft, Writing – review & editing, Project administration, Formal analysis.

- **Carl Hvarfner**: Investigation, Formal analysis, Software, Writing – review & editing, Validation.

- **Samuel Müller**: Methodology, Formal analysis, Writing – review & editing, Software, Validation.

- **Frank Hutter**: Funding acquisition, Supervision, Resources, Project administration.

- **Eytan Bakshy**: Supervision, Writing – review & editing, Project administration.

---

[3]See https://credit.niso.org/.

