# $\alpha$-PFN: Fast Entropy Search via In-Context Learning

## Abstract

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

206 Recently, (Hu et al., 2024) proposed a transformer called
207 InfoNet for amortizing computations regarding mutual in-
208 formation. Our work is similar, since acquisition functions
209 for entropy search are also defined in terms of mutual in-
210 formation. However, the second stage amortizing using our
211 $\alpha$-PFN offers even more speed-up compared to applying
212 InfoNet for computing acquisitions, as we don't need to
213 approximate the expectation over optima. Surrogate models
214 can also find applications in bandits, see for example (Zhou
215 et al., 2020).

## 3. Entropy Search with PFNs

In the following we explain how we train $\alpha$-PFN, which pre-
dicts the acquisition values for PES, MES, and JES directly.
See Figure 2 for an overview of the training setup. To gen-
erate training data for the $\alpha$-PFN, we first train an auxiliary
(base) PFN. The base PFN is trained to make $y$ predic-
tions (optionally) conditioned on information of the location
and/or value of the optimum. Next to the ordinary PPD
$p(y|D_{trn}, x)$, this base PFN can compute $p(y|D_{trn}, x, I)$
with $I = x^*$ for PES, $I = f^*$ for MES, and $I = (x^*, f^*)$
for JES, as illustrated in Figure 1. For each ES variant, we
now train a second PFN model, the $\alpha$-PFN, that directly
predicts the acquisition function. More specifically, $\alpha$-PFN
is trained to approximate the full information gain distribu-
tion by using the base PFN's predictions, conditioned on
the datasets' true optimum (precomputed, see below), as
training targets, and at test time we recover the acquisition
function (expected information gain), as the mean thereof.

**Pre-computing Gaussian Process prior data.** To con-
struct our PFN models for GP inference we need to train
the model on millions of dataset samples from a GP prior.
Furthermore, we need to know $x^*$ and $f^*$ for each dataset,
which is not feasible to compute for an exact GP sample. To
make this feasible, we approximate the GP samples using
Random Fourier Features (RFFs; Rahimi & Recht, 2007).
The results of this precomputation are (approximate) sam-
ples, that can be queried efficiently for arbitrary $x$, on which
we can employ gradient-based optimization to find approxi-
mate $x^*$ and $f^*$. See Appendix B for more details.

**Training the base PFN.** When we are training on our
precomputed RFF GP data, we have access to $x^*$ and $f^*$,
and feed these to the PFN to condition on them. We add
one extra data point to the context of the PFN, which is
encoded just like the other data points but using a different
encoder, so that the PFN will learn to treat it differently. We
randomly pass $x^*$ and $f^*$ with a $50\%$ chance such that a
single model is able to handle all four cases equally. This
way we train our base PFN $q(y|x, D_{trn}, I)$, where $I$ can
contain $x^*$ and/or $f^*$. For more details see Appendix C.2.

**Training $\alpha$-PFN.** After training the base model, we train
a second PFN model, the $\alpha$-PFN, that takes the observation
data $D$ and query point $x$ as input and predicts the acquisi-
tion $\alpha(x, D)$ directly. We train $\alpha$-PFN to use $D_{trn}$ and $x$ to
predict

$$H(q(y|D_{trn}, x)) - H(q(y|D_{trn}, x, I)), \qquad (5)$$

where $I$ are extra information on the optimum. The (dif-
ferential) entropy is computed analytically on the PFNs

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

 90M pre-generated GP sample paths (Appendix B) from the hyperprior with varying dimensionalities (1-6D). We use varying lengthscales per dimensionality, e.g. with automatic relevance determination. For all details regarding the prior and model training, see Appendix C.

## 5. Experimental Setup

The goal of the experiments is to show that $\alpha$-PFN is a practical and efficient alternative to GP-based approximations of Entropy Search. For this reason, we focus on a specific setting where PFN and GP use *the same exact* prior (defined in Appendix C.1), such that we can compare performances and runtimes of the two approaches directly. However note that sharing the same prior also implies that PFN and GP should exhibit similar behavior and are not expected to show a large performance difference. Specifically, we do not expect GP-ES to be state-of-the-art on these benchmarks, so we do not claim $\alpha$-PFN variants to be either. Our evaluation stress tests $\alpha$-PFN. Most of the functions under evaluation will not match this restrictive prior, i.e., we evaluate $\alpha$-PFN out-of-distribution. We also test the extrapolation capabilities of $\alpha$-PFN to generalize on large dimensions (up to 16D) and context sizes (100 BO iterations), while being trained on smaller datasets (up to 6D) and context sizes (up to 50).

**Baselines.** We compare $\alpha$-PFN against existing Entropy Search approximations in the BoTorch library (Balandat et al., 2020): JES (Hvarfner et al., 2022), (GIBBON-)MES (Moss et al., 2021) and PES (Hernández-Lobato et al., 2014). Since no fully Bayesian entropy search implementations exist, we compare against MCMC-ES instead, which approximates a posterior over the acquisition, rather than computing the acquisition of the fully Bayesian model. For our baselines we use NUTS (Hoffman et al., 2014), which uses Hamiltonian Monte Carlo (HMC), again making use of the BoTorch library for JES, GIBBON-MES and PES. Furthermore, we include EI as a reference, to show the merits of using entropy-based acquisition functions.

**Evaluation datasets.** To show robustness to out-of-distribution, we evaluate on well-known synthetic test functions for black-box optimization, including synthetic functions (Branin, Hartman, Ackley) and real HPO tasks (LCBench (Zimmer et al., 2021), HPO-B (Pineda-Arango et al., 2021a)). Test functions include both continuous and discrete domains.

Further details regarding the evaluation protocol can be found in Appendix D.

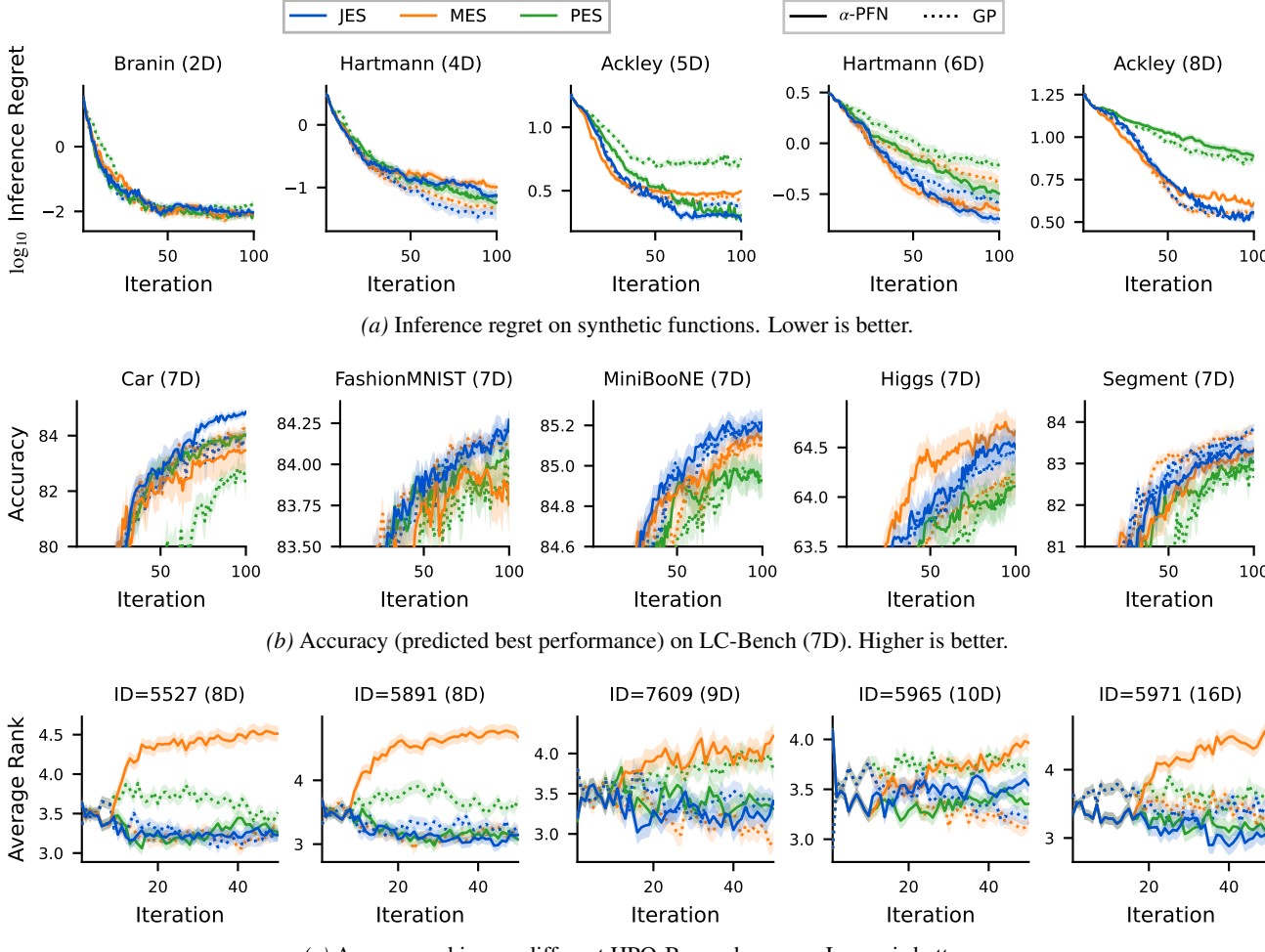

*(a)* Inference regret on synthetic functions. Lower is better.

*(b)* Accuracy (predicted best performance) on LC-Bench (7D). Higher is better.

*(c)* Average ranking on different HPO-B search spaces. Lower is better.

*Figure 3.* Bayesian optimization performance comparison between GP-MCMC (NUTS) and α-PFN across different synthetic test functions and real HPO benchmarks. The shaded area indicates a standard error.

## 6. Results

For clarity, we include only regret-based comparison of JES, MES, and PES. Additional results, including comparison to EI (Figure 4), and pair-wise win-rate (Figure 5) are available in Appendix G. Note that, while EI, especially on HPOB, yields strong performance, our goal is to compare across the ES methods. We also show in Appendix G that the EI PFN is competitive with the GP EI.

**Synthetic Test Functions.** The results are in Figure 3a, which are the result of 30 repetitions. The PFN often closely matches the GP performance. This is also supported by the moderate win-rates in Figure 5. In terms of average regret, PES variants are generally competitive or superior. The JES variant under-performs on Hartmann 4D, so does the MES variant, which also shows degraded performance on large context sizes on Ackley 8D. On the other hand, all PFN variants perform better on Hartmann 6D, and JES

additionally on Ackley 5D.

**Real-World HPO Tasks (LCBench, HPO-B).** In our final experiment, we consider optimizing real-world black box optimization tasks, see Figures 3b-3c. Results are from 30 randomly initialized runs. Performances are often quite close, which is also reflected in the average rankings (over problems) that are quite high (last panel). On LCbench, α-PFN variants often outperform the baselines, except on Segment. Overall, the JES-α-PFN performs consistently better across all tasks considered. While MES-α-PFN performs best in Higgs (LCbench), its performance is worse, often outperformed by the GP baseline, in particular on HPO-B. We will investigate this further as future work.

**Timing Results.** Table 1 reports the median cumulative runtime (in minutes) for GP-MCMC and α-PFN across all benchmarks. α-PFN delivers substantial speedups across all tasks and acquisition functions, consistently outperforming

*Table 1.* Runtime comparison (minutes) between GP-MCMC and $\alpha$-PFN across tasks. Speedup = GP/$\alpha$-PFN. Values represent the median cumulative runtime of model fitting and acquisition optimization across BO iterations. Domain: C = Continuous, D = Discrete.

| | Branin | Hartmann | Ackley | Hartmann | Car | Fashion | Higgs | MiniBooNE | Segment | Ackley | HPOB-5527 | HPOB-5891 | HPOB-7609 | HPOB-5965 | HPOB-5971 |
|---|---|---|---|---|---|---|---|---|---|---|---|---|---|---|---|
| Dim | 2 | 4 | 5 | 6 | 7 | 7 | 7 | 7 | 7 | 8 | 8 | 8 | 9 | 10 | 16 |
| Domain | C | C | C | C | C | C | C | C | C | C | D | D | D | D | D |
| **GP Fully Bayesian** | | | | | | | | | | | | | | | |
| MES | 18.3 | 57.0 | 30.9 | 81.6 | 180.6 | 125.4 | 80.1 | 223.1 | 48.8 | 45.5 | 58.0 | 51.8 | 74.5 | 122.3 | 69.4 |
| JES | 26.8 | 55.0 | 60.4 | 172.3 | 259.7 | 123.4 | 102.7 | 150.0 | 66.8 | 78.9 | 75.9 | 82.6 | 41.5 | 128.1 | 86.2 |
| PES | 34.3 | 95.0 | 78.0 | 117.7 | 213.7 | 169.5 | 94.6 | 107.7 | 104.2 | 79.0 | 63.8 | 97.0 | 100.2 | 193.9 | 92.9 |
| **$\alpha$-PFN** | | | | | | | | | | | | | | | |
| MES | 5.4 | 11.1 | 12.6 | 18.9 | 21.0 | 16.7 | 28.1 | 28.4 | 30.0 | 23.4 | 10.5 | 1.7 | 1.1 | 10.1 | 2.9 |
| JES | 6.1 | 11.9 | 14.9 | 18.9 | 19.9 | 25.1 | 32.5 | 19.7 | 32.8 | 23.5 | 8.4 | 1.4 | 1.2 | 7.7 | 2.3 |
| PES | 5.3 | 9.3 | 15.5 | 20.3 | 20.9 | 15.0 | 19.6 | 26.1 | 25.4 | 20.6 | 11.7 | 1.7 | 1.4 | 9.4 | 2.9 |
| **$\alpha$-PFN Speedup ($\times$)** | | | | | | | | | | | | | | | |
| MES | **3.4** | **5.1** | **2.4** | **4.3** | **8.6** | **7.5** | **2.9** | **7.9** | **1.6** | **1.9** | **5.5** | **31.3** | **65.0** | **12.1** | **24.2** |
| JES | **4.4** | **4.6** | **4.0** | **9.1** | **13.1** | **4.9** | **3.2** | **7.6** | **2.0** | **3.4** | **9.1** | **58.7** | **34.4** | **16.7** | **37.6** |
| PES | **6.4** | **10.2** | **5.0** | **5.8** | **10.2** | **11.3** | **4.8** | **4.1** | **4.1** | **3.8** | **5.5** | **57.2** | **72.4** | **20.6** | **31.5** |

GP-MCMC in computational efficiency. Speedups range from $1.6\times$ to over $72\times$, with particularly important gains on HPO-B tasks where $\alpha$-PFN achieves speedups exceeding $30\times$–$70\times$. These results demonstrate that $\alpha$-PFN not only matches the optimization performance of GP-based entropy search methods but does so at a much at a lower computational cost, making $\alpha$-PFN practical for larger-scale applications.

## 7. Summary, limitations, and future research

Our results demonstrate that PFNs can be used for Entropy Search. We show that $\alpha$-PFN is capable of simulating the state-of-the-art (JES) at reduced runtimes. The strong speed improvements highlight its ability to learn more efficient approximations than handcrafted alternatives. Note that, it is difficult to compare runtimes, as the hyperparameters of the baselines, such as the number of MC samples used, can influence the runtimes significantly. We tried to set these to reasonable values.

The $\alpha$-PFN often matches performance of the GPs, except for functions or datasets that are out of distribution. One way to mitigate this, is through developing more diverse priors or transformations at test time. A major strength of our framework is its flexibility. While we used a GP-based hyperprior to align with standard ES methods, the PFN approach is agnostic to the choice of prior, e.g., Bayesian neural networks, ensembles, etc. could be used easily. Exploring alternative priors is a promising direction for future work. One bottleneck is that the $\alpha$-PFN needs to be re-trained for each prior; something that may be resolved by methods like Whittle et al. (2025). So far, we have trained

models up to 6 dimensions for 50 iterations, and find that the PFNs can generalize to higher dimensional problems or higher iterations. The work of Yu et al. (2025) illustrates that our architecture should in principle be scalable to high-dimensional BO (up to 500D). However, this is a significant compute

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

 from an exponentially-biased distribution favoring shorter contexts, with probabilities proportional to $0.99^c$ for $c \in [0, \; 50 \cdot D - 1]$. The context and query points are chosen as described in Appendix E.

We closely follow the original PFN architecture and training pipeline, used in previous works (Müller et al., 2021; 2023). We use a small ($\pm$15M parameter) decoder-only transformer, with 6 layers, each using an embedding size of 128, 4 attention heads, and 1024 units in the hidden expansion layer. We use the PFN regression head proposed by Müller et al. (2021) to model the output distribution. The output distribution is the Full-Bar Distribution (also called Riemannian distribution) with Full-Support. We use 5000 bins and determine the bin size such that each bin contains roughly as many of the training targets.

### C.1. Training Prior

We train our models using a Fully Bayesian GP prior. The input dimension $d$ ranges from 1 to 6. We use the squared exponential kernel with an output scale $\sigma_f = 1$. The lengthscale per dimension is sampled independently (corresponding to Automatic Relevance Determination) from $P(D) = LN(\mu_0 + \frac{\log D}{2}, \sigma_0)$ with $\mu_0 = -0.75$ and $\sigma_0 = 0.75$ (this prior was inspired by Hvarfner et al. (2024)). We add zero mean Gaussian noise with standard deviation $\sigma_n$ sampled from $LN(-4, 1)$. The mean $\mu$ of the GP is sampled from $N(0, 0.5)$ per task.

### C.2. Base PFN Training Details

Recall that the base model is trained to condition on $x^*$, $f^*$ or both. During training, using pregenerated $x^*$ and $f^*$ values from GP sample paths (Appendix B), we randomly sample the conditioning to train the base model: (1) no conditioning, (2) condition on $x^*$, (3) condition on $f^*$, (4) condition on both. We train a single base model to support all four cases. These cases are all equally probable during training. Overall, accounting for the data augmentation, we train our base model on

300M datasets.

### C.3. $\alpha$-PFN Training Details

The PFN model directly predicting the (expected) information gain closely follows the architecture and training of the base model. The main difference is that no conditioning tokens were used (and it does not take $I$ as input), and that the prediction targets for queries are not $y$, but the oracle information gain from equation 5. Note that we trained a $\alpha$-PFN for each ES variant. Each $\alpha$-PFN is trained on 300M datasets, using similar data augmentation as the base model. Furthermore, we use the full-support output head (Müller et al., 2021), where the outmost "bins" are open and modeled as half-normals.

## D. Bayesian Optimization experiment details

The initial design consists of $d$ uniformly sampled points. At each BO iteration, the acquisition function is optimized using Botorch routine `optimize_acqf`, with the following hyperparameters: 128 uniform points as initial candidates and 10 restarts. We follow a similar optimization procedure to compute the maximizer of predictive posterior distribution, required for computing the inference regret.

We evaluate all methods in terms of inference regret for synthetic tasks, which is the standard evaluation measure for methods using information theoretic acquisition functions. Inference regret is defined as $f(x^*) - f(\hat{x}^*)$, where $\hat{x}^*$ is the maximizer of the posterior predictive distribution, i.e., $\hat{x}^* = \text{argmax}_{x \in A} \mathbb{E}[q(y|D, x)]$. Note that this maximizer is approximated by performing gradient descent on $q(y|D, x)$, which in this case can either be the surrogate GP or the PFN base-model, with a setup similar to that considered for the acquisition function optimization. For acquisition function optimization we use the standard Botorch routine for GPs, but for the PFN we deviate from this and use a new simpler optimizer. For real-world tasks we report $f(\hat{x}^*)$ (similar to inference regret). For evaluation we use 100 iterations, while PFNs are trained for 50.

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

*(c)* HPO-B.

*Figure 5.* Win rate comparison across different benchmarks.