# OpenReview forum: "$\alpha$-PFN: Fast Entropy Search via In-Context Learning"
_ICML.cc/2026/Conference — ICML 2026 regular_

### Official Review · Reviewer_mryg · 2026-03-02

**Soundness:** 3
**Presentation:** 4
**Significance:** 3
**Originality:** 4
**Overall Recommendation:** 5
**Confidence:** 4

**Summary:**

The authors propose a doubly amortised approach to entropy-search-based Bayesian optimisation. Firstly, they use a PFN instead of a GP as the probabilistic surrogate model of the black-box function. As in previous work, this facilitates a fully Bayesian procedure in which no nuisance parameters (such as kernel lengthscales) need to be point-estimated. Secondly, they introduce a further PFN, $\alpha$-PFN, which is trained to predict an entropy-search acquisition function directly. In using $\alpha$-PFN, the authors avoid the need for tedious and stochastic Monte Carlo estimation of the acquisition function, and the cost of evaluating many such acquisition functions is amortised across the meta-style pretraining phase. The resulting approach positions the combination of entropy-search and fully Bayesian surrogates as a pragmatic and effective tool in the Bayesian optimiser's toolbox.

**Compliance With Llm Reviewing Policy:**

Affirmed.

**Final Justification:**

I only had rather minor/editorial suggestions in my review, and these have all been addressed by the authors in their response. This paper would make a valuable contribution to the ICML community, and I hope to see it accepted!

**Key Questions For Authors:**

I have none for now.

**Limitations:**

yes

**Strengths And Weaknesses:**

### Strengths:

- The writing and exposition is of very high quality and I think the manuscript leaves no room for misunderstandings caused by insufficient clarity. Figure 2 is simply excellent.

- The idea is solid and it represents a respectable contribution for the BayesOpt community. It is exciting to see such creative uses of amortised inference.

- Since the method enables practitioners to avoid noisy and slow Monte-Carlo-based acquisition function optimisation, the results only need to demonstrate some approximate level of parity between $\alpha$-PFN and traditional methods in terms of performance, and they do just that. The significant speedups are where the real value of $\alpha$-PFN lies. For the AC's sake, it could be worth pointing out here that, to complain about $\alpha$-PFN "not winning" in every single case (performance-wise) is to miss the point.

- There are sufficient details to reproduce the experimental setup.

- The theoretical results are sound.

### Weaknesses:

1. In my opinion, the presentation of "Insight 1" is the weakest expository part of the paper. An "insight" is not a formal mathematical statement type in the way that a proposition, lemma, corollary, or theorem is. My recommendation would be to change it into a proposition, and rather use a statement along the lines of "*Minimising the objective* $l_\theta$ *is equivalent to minimising the task-averaged KL divergence between* $p(\tilde{\alpha}|D, x)$ *and the* $\alpha$*-PFN’s output. Evaluating the mean of a trained* $\alpha$*-PFN's output then directly approximates the corresponding ES acquisition function value.*". This wording is only a suggestion; feel free to reword as desired.

2. Recently there has been a flurry (e.g. [1](https://openreview.net/forum?id=5uQL7oVAX8), [2](https://proceedings.mlr.press/v235/hvarfner24a.html), [3](https://openreview.net/forum?id=kX8h23UG6v)) of papers essentially saying "vanilla GP + LogEI/UCB is all you need for BayesOpt". While I am of the view that increasing the variety of tools available to practitioners is motivation enough, some readers might wonder why we should bother with ES at all if (Log)EI and UCB already seem to work well. In the next revision, perhaps the authors can provide some more compelling arguments for why developing tractable ES-based approaches is something readers should care about.

### Nit-Picks:
(These are not criticisms, the purpose is to help the authors in tidying up the manuscript for revision)

a.) 6th line of last paragraph of section 1, "$\alpha$-PFNtransformer" should be "$\alpha$-PFN transformer" (i.e., there is a missing space between "PFN" and "transformer").

b.) 10th line of first paragraph of section 2.1, I think "transformer" should be replaced with "PFN", "TNP", or "transformer-based probabilistic model". A transformer is not an example of a general-purpose Bayesian surrogate model.

c.) In the final line of section 2.1, I think there is a stray $y_{t+1}$ after the full stop. It should be removed.

d.) The penultimate sentence of section 2.2 does not make sense. Perhaps "would consists of superposition" can be replaced with "would consist of a superposition"?

e.) In the 4th line of the second paragraph of section 2.4, should "condition on $f^\*$ and predict $f^\*$" instead be "condition on $D_\text{trn}$ and predict $f^\*$"?

f.) In the 5th line of the third paragraph of section 2.4, "amortizing" should instead be "of amortization".

g.) In the 3rd line of the first paragraph of page 5, "info" should rather be "information".

h.) In the final paragraph of section 4, the authors mention automatic relevance determination (ARD). To my knowledge, ARD refers to the optimisation of the marginal likelihood w.r.t. feature importance/inverse lengthscale hyperparameters. As far as I can tell, this is *not* what is being done in the PFN. Rather, the training procedure necessitates a hyperprior over such lengthscales, and the PFN amortises inference over them (i.e., these hyperparameters are *marginalised* rather than *optimised*).

i.) In the final paragraph before the Impact Statement, the phrase "acquisition function amortization" is imprecise. It makes no sense to *amortise a function*. It does make sense to amortise the cost of evaluating many such functions. My recommendation would be to rather use the phrase "amortization of acquisition function evaluation"

j.) There are generally far too many commas. In the second paragraph of section 2.3, there should be no comma after "The versatility of this paradigm". In the 3rd line of the third paragraph of section 2.4, there should be no comma after "Our work is similar". In the first paragraph of section 6 there are far too many, particularly in the third sentence. There should be no comma after "Note that" in the first paragraph of section 7. The second paragraph of section 7 also has very many commas; some are not wrong but also not needed, some are definitely wrong (e.g., after "One way to mitigate this").


I hope the authors find my review to be helpful, and I commend them on their excellent work. I look forward to discussing things further should they so wish!

---

> ### Author Rebuttal · Authors · 2026-03-31
>
> Thank you for the constructive review and for the very concrete suggestions on both exposition and positioning. In the revision we will turn Insight 1 into a proposition, improve the motivation for ES, and apply your wording and punctuation fixes.
>
> **Insight 1 should be a formal statement**
>
> We will follow your suggestion here - thanks a lot for the detailed feedback.
>
> **Why bother with ES if vanilla GP plus LogEI or UCB works well**
>
> Valid question. We agree that these methods often work well. We are not claiming ES always beats LogEI or UCB. Our view is that ES is under-evaluated mainly because it is expensive and brittle to implement, so results are hard to reproduce and compare across papers, e.g. [1] vs [4], [2] vs [3]. Prior work [1, 2] has shown settings where ES-style objectives can be extremely effective, including inference regret, high noise, and multi-objective regimes. Our contribution makes entropy search easier and cheaper, facilitating broader adoption, in practical applications, but also as a scientific baseline - encouraging a more thorough evaluation.
>
>
>
> **Nitpicks and clarity improvements**
>
> We will incorporate your suggestions for the revised version. Indeed, we use an ARD-kernel, where we marginalize the hyperparameters out (lengthscale per dimension is marginalized out). Thanks for all the very useful comments!
>
> [1] Carl Hvarfner, Frank Hutter, Luigi Nardi. Joint Entropy Search for Maximally-Informed Bayesian Optimization. NeurIPS 2022.
>
> [2] Ben Tu, Axel Gandy, Nikolas Kantas, Behrang Shafei. Joint Entropy Search for Multi-Objective Bayesian Optimization. NeurIPS 2022.
>
> [3] Sam Daulton, Maximilian Balandat, Eytan Bakshy. Hypervolume Knowledge Gradient: A Lookahead Approach for Multi-Objective Bayesian Optimization with Partial Information. ICML 2023.
>
> [4] Sebastian Ament, Sam Daulton, David Eriksson, Maximilian Balandat, Eytan Bakshy. Unexpected Improvements to Expected Improvement for Bayesian Optimization. NeurIPS 2023.

---

> > ### Author Rebuttal · Reviewer_mryg · 2026-04-01
> >
> > Many thanks for the nice rebuttal. If you make the changes you promise to do so, then all of my concerns will have been addressed. Furthermore, thanks for helping me to understand the details of your ARD kernel as well as for discussing your motivation behind ES in more detail.
> >
> > I maintain my favourable score, and hope to see the paper accepted!

---

> > > ### Author Response · Authors · 2026-04-07
> > >
> > > Thanks for your time and the positive assessment of our work.

---

### Official Review · Reviewer_Ygzd · 2026-03-09

**Soundness:** 2
**Presentation:** 2
**Significance:** 2
**Originality:** 3
**Overall Recommendation:** 2
**Confidence:** 4

**Summary:**

This paper proposes a fast, learned approximation to information-theoretic acquisition functions for Bayesian optimization, focusing on Entropy Search (ES) variants whose practical use is often limited by expensive Monte Carlo estimates of information gain and specialized implementations. The method introduces a two-stage amortization strategy based on Prior-data Fitted Networks (PFNs): a first PFN models the predictive distribution while being conditioned on optimum-related information, and a second model (the $\alpha$-PFN) is trained to predict the expected information gain so that ES-style acquisitions can be evaluated via a single forward pass per candidate point. Across synthetic functions and real-world hyperparameter-optimization benchmarks, the approach aims to match the optimization performance of standard ES implementations while substantially reducing acquisition-evaluation cost, reporting large speedups in practice.

**Compliance With Llm Reviewing Policy:**

Affirmed.

**Key Questions For Authors:**

See weaknesses.

**Limitations:**

See weaknesses.

**Strengths And Weaknesses:**

Strengths
1. This paper ddresses a well-known bottleneck of ES-type methods---slow and implementation-heavy information-gain approximations---with a simple inference-time interface (one forward pass per candidate).
2. Separating (i) modeling/predicting the relevant posterior quantities from (ii) learning the information-gain mapping yields a modular and reusable pipeline.
3. Once trained, the method replaces repeated Monte Carlo computations with deterministic neural inference, which is attractive for inner-loop BO workloads.
4. The evaluation includes both controlled synthetic benchmarks and more realistic hyperparameter-optimization settings, which helps demonstrate applicability beyond toy cases.

Weaknesses
1. The PFNs are pretrained on synthetic datasets drawn from a particular GP prior/hyperprior family; if the real BO task exhibits different smoothness, kernel structure, noise characteristics (e.g., heavy-tailed/heteroscedastic), or nonstationarity, the learned acquisition may become miscalibrated and select systematically suboptimal queries.
2. Training requires generating massive amounts of prior-sampled data and computing information-gain targets, including approximate optima (e.g., via many GP sample paths / random Fourier feature sampling). This shifts complexity from test-time to a substantial offline pipeline that may be difficult to reproduce and expensive for practitioners without significant compute.
3. The approach trains separate $\alpha$-PFNs for different ES variants (e.g., PES/MES/JES). In practice, changing the acquisition definition, kernel family, input dimensionality regime, or dataset normalization can require retraining, reducing the ``plug-and-play'' appeal compared to standard ES implementations.
4. The method learns an approximation to expected information gain; however, the paper provides limited quantitative analysis of (i) calibration/accuracy of the predicted information gain, (ii) how approximation error varies across the candidate set, and (iii) whether systematic bias in the learned acquisition can lead to repeated selection of similar points (mode-seeking) or premature exploitation.
5. The evaluation demonstrates strong speedups on the chosen synthetic and HPO benchmarks, but it is less clear how the method behaves in high dimensions, with large BO budgets, with constraints/multi-fidelity objectives, or under expensive-to-evaluate models where acquisition optimization itself becomes the dominant cost.

---

> ### Author Rebuttal · Authors · 2026-03-31
>
> Thank you for your critical assessment. We do not disagree with your main arguments, and we will include a discussion of the weaknesses you raised in our revision. That said, most (1,2,3,5) apply to amortized inference in general, are not specific to our approach and beyond its envisioned scope. We agree these are important and that future work addressing these practical concerns is crucial to facilitate broader adoption, but this goes well beyond the envisioned scope of our work. In our specific setting (ES amortization), quantitative analysis of approximation quality (4) is complicated by a lack of a ground-truth / consistent estimator. That said, we provide an illustration ([See Rebuttal PDF, Figure 1](https://figshare.com/s/804612ec2ba937e581b7)), comparing our amortized (alpha-PFN) to different MC-based approximations of entropy search.
>
> **Pretraining on a particular GP prior may lead to miscalibration under prior mismatch**
>
> Prior mismatch matters, but it is primarily a model-level robustness issue rather than an acquisition-function issue. That said, we added a stress test with extreme mismatch by using a very large noise level 0.5 where the signal scale of the benchmark is less than 1. This noise level is extremely unlikely under our prior. Under this shift, performance degrades at roughly the same rate as the GP baseline, which naturally faces the same mismatch issue.  Please also note that while our model was trained up to 6D, it generalizes to higher dimensions (up to 16D on HPO-B, see Figure 3). If you have a specific additional mismatch experiment you want, we will try to accommodate it.
>
> Also, our approach is compatible with broader and more exotic priors, including the HEBO [1] prior used in PFNs4BO and sparse priors such as SAAS [2].
>
> [1] Muller et al PFNs4BO Prior Data Fitted Networks for Bayesian Optimization ICML 2023
> [2] Eriksson and Jankowiak High Dimensional Bayesian Optimization with Sparse Axis Aligned Subspaces UAI 2021
>
> **The offline pipeline may be expensive and hard to reproduce**
>
> The training costs are detailed in Appendix F. In addition to model checkpoints, we will provide a complete pipeline (data and code) to fully reproduce our model upon acceptance.
>
> **Training separate models per ES variant reduces plug and play appeal**
>
> We agree that training one model per acquisition may be inconvenient, but it is not the (only) intended usage pattern in practice. One could envision two modes. First, a family of small, efficient, specialized models covering the most common cases (or pretrained by the user for their custom setting). Second, a single (foundation) model pretrained on a wide variety of data, that supports conditioning of specific sub-classes (e.g., using a specific hyper-prior, kernel, etc.). The PFN architecture naturally supports such conditioning, e.g., following a similar approach as we used in our base PFN model to support the 4 conditional prediction scenarios. That said, creating such a foundation model is a non-trivial feat, and subject to future work. We will clarify this framing.
>
> **Limited quantitative analysis of information gain accuracy**
>
> We agree that our work, beyond its theoretical argument and extensive practical BO evaluation, would benefit from a dedicated quantitative analysis of the approximation itself. Unfortunately, to the best of our knowledge, no ground truth or even consistent estimator for entropy search exists. Lacking such reference, we are unsure what quantitative analysis we can meaningfully perform. If you have a specific analysis in mind, let us know and we will try to accommodate it. As an illustration, we provide a qualitative comparison of alpha-PFN to different MC-based approximations of entropy search ([See Rebuttal PDF, Figure 1](https://figshare.com/s/804612ec2ba937e581b7)).
>
> **Unclear behavior in high dimensions and broader regimes**
>
> This paper presents a way to amortize acquisition evaluation, not a method whose goal is robustness to mismatch or a solution to high-dimensional / constrained / multi-fidelity BO. We used moderate-dimensional, practically relevant benchmarks and hope you see this as the right scope. Higher-dimensional regimes can require larger budgets for amortization to pay off, and modeling improvements for high-D and robustness are compatible with PFNs (see [4] who built a PFN up to 2000 dimensions and 100K data points). Our current work already uses a dimension-scaling prior in the spirit of [3], and we will clarify this point and keep broader high-D claims as future work.
>
> [3] Hvarfner, Hellsten, Nardi. Vanilla Bayesian Optimization Performs Great in High Dimensions. ICML 2024.
>
> [4] Grinsztajn, L., Flöge, K., Key, O., Birkel, F., Jund, P., Roof, B., ... & Hutter, F. (2025). TabPFN-2.5: Advancing the State of the Art in Tabular Foundation Models. arXiv:2511.08667.
>
> ________
>
> We hope you find these responses satisfactory. If you have additional questions or concerns, please do not hesitate to raise them.

---

> > ### Author Rebuttal · Reviewer_Ygzd · 2026-04-04
> >
> > Thank you for your response. I will maintain my score.

---

> > > ### Author Response · Authors · 2026-04-07
> > >
> > > Thank you for your response and your time. We appreciate any further feedback that you think remains unaddressed. We will incorporate it to improve our work further.

---

### Official Review · Reviewer_NsHj · 2026-03-11

**Soundness:** 2
**Presentation:** 2
**Significance:** 3
**Originality:** 3
**Overall Recommendation:** 4
**Confidence:** 2

**Summary:**

The first PFN models predictive distributions conditioned on information about the optimum; the second PFN learns the distribution of information gain and uses its mean as a fast approximation to PES, MES, and JES. The paper positions this as a replacement for Monte Carlo-heavy entropy search approximations, and evaluates it on synthetic BO tasks and real HPO benchmarks.

**Compliance With Llm Reviewing Policy:**

Affirmed.

**Final Justification:**

I think it is a good paper, but I am not entirely certain

**Key Questions For Authors:**

a. How sensitive is performance to prior mismatch, the RFF approximation quality, and the BO-trace clustering heuristic used during training?

b. Can the authors quantify the one-time pretraining cost and the amortization threshold?

**Limitations:**

see weakness

**Strengths And Weaknesses:**

Strengths :

The paper addresses a meaningful problem. Information-theoretic acquisition functions are attractive but often too slow in practice, so amortizing their computation is valuable.

The core idea is novel and well-motivated. The separation into a conditional base PFN and an acquisition PFN is conceptually clean.

The method is fairly general within the entropy-search family, covering PES, MES, and JES under one framework.

Weaknesses

a. The paper does not sufficiently isolate where the gains come from. A key missing ablation is a comparison against base PFN + Monte Carlo over optima at test time. Without this, it is hard to quantify the added value of the second amortization stage.

b. The claim regarding the fully Bayesian advantage is interesting, but not yet fully demonstrated empirically. The paper argues that $\alpha$-PFN can better incorporate hyperparameter uncertainty, but I did not see a dedicated ablation that clearly validates this claim.

---

> ### Author Rebuttal · Authors · 2026-03-31
>
> Thank you for your review. In our response, we include a qualitative comparison of the alpha-PFN predictions against the suggested MC-based baselines ([Rebuttal PDF, Figure 1](https://figshare.com/s/804612ec2ba937e581b7)). We also include additional results, demonstrating robustness against prior mismatch ([Rebuttal PDF, Figure 3](https://figshare.com/s/804612ec2ba937e581b7)) and the importance of our BO-trace clustering heuristic ([See Rebuttal PDF, Figure 2](https://figshare.com/s/804612ec2ba937e581b7)). We will also include these figures in our revision.
>
> **The paper does not isolate where the gains come from and base PFN plus Monte Carlo is missing**
>
> Thank you for suggesting this baseline. We have included a qualitative plot comparing JES-PFN against MC approaches using base-PFN and GP-based approximations. This illustration ([Rebuttal PDF, Figure 1](https://figshare.com/s/804612ec2ba937e581b7)) shows that while acquisition peaks are roughly aligned, the conditional base-PFN provides “richer” information gain estimates. It also shows that MC-based estimates (for 10 different seeds, each based on 128 optima samples) are highly variable in regions of interest, noise that alpha-PFN is pretrained to model and average out. In this 1D setting, with fixed GP hyperparameters, PFN-based approaches provide little speed up. However, in the fully Bayesian setting, especially in higher dimensions, the process used to generate MC samples becomes the main computational bottleneck - which alpha-PFN avoids entirely.
>
>
> **The fully Bayesian advantage is interesting but not yet empirically demonstrated**
>
> Our approach uniquely enables a fully Bayesian treatment of entropy search. Conceptually pleasing as this may be, we found isolating the practical benefit/effect to be non-trivial. We chose “fully Bayesian” GP baselines for a fair comparison with $\alpha$-PFN, our work is not intended to demonstrate the superiority of a fully Bayesian approach. We will adjust wording to avoid overstating the claim.
>
> **How sensitive is performance to prior mismatch**
>
> Our view is that prior mismatch is primarily a model-level robustness issue, and we see the acquisition approximation as orthogonal to that. That said, we added a stress test ([Rebuttal PDF, Figure 3](https://figshare.com/s/804612ec2ba937e581b7)) with extreme mismatch by using a very large noise level 0.5 where the signal scale of the benchmark is less than 1. This noise level is extremely unlikely under our prior. Under this shift, performance degrades at roughly the same rate as the GP baseline, which naturally faces the same mismatch issue.  Please also note that while our model was trained up to 6D, it generalizes to higher dimensions (up to 16D on HPO-B, see Figure 3). If you have a specific additional mismatch experiment you want, we will try to accommodate it.
>
> More generally, our method is compatible with broader or more exotic priors, including the HEBO prior used in PFNs4BO [1] and sparse priors such as SAAS [2]. Such advanced priors provide a clear path to reducing prior mismatch for future work.
>
> [1] Muller et al PFNs4BO Prior Data Fitted Networks for Bayesian Optimization ICML 2023
> [2] Eriksson and Jankowiak High Dimensional Bayesian Optimization with Sparse Axis Aligned Subspaces UAI 2021
>
> **How sensitive is performance to the RFF approximation quality**
>
> We did not systematically vary the number of random Fourier features. We follow the same default feature count used by the ES acquisition functions we evaluate, using 1024 features, which worked well empirically in our setting. Moreover, we used the variance-reducing approach of [3] as is convention in BoTorch.
>
> [3] Sutherland, Danica J and Jeff Schneider On the Error of Random Fourier Features Advances in Neural Information Processing Systems 2015
>
> **Sensitivity to BO-trace clustering heuristic**
> We performed an ablation of our BO-trace clustering heuristic, contrasting it with the standard uniform training input data ([See Rebuttal PDF, Figure 2](https://figshare.com/s/804612ec2ba937e581b7)). These results show that while the standard approach is competitive at lower dimensions, for PES, it does not scale up (beyond 4D, MES/JES).
>
> **Can you quantify the one-time pretraining cost and the amortization threshold**
>
> The one-time compute cost pretraining is detailed in Appendix F (base: 70 hours, alpha: 12 hours).  The exact amortization threshold depends strongly on the application (e.g., dimensionality of target function). At the risk of over-simplifying, 100 runs on Ackley 8D would roughly amortize these costs.
>
> ________
>
> We hope you find these responses satisfactory. If you have additional questions or concerns, please do not hesitate to raise them.

---

> > ### Author Rebuttal · Reviewer_NsHj · 2026-04-01
> >
> > First of all, I would like to sincerely apologize for mistakenly raising the Ethics Flag. I realize this may have been alarming or upsetting to the authors, and I am truly sorry for this oversight. To be clear, there are no ethical issues with this paper. If there is a mechanism to retract the flag, I would like to do so immediately. I also kindly request the Area Chair (AC) to note that this was purely a mistake on my part.
> >
> > Aside from my mistake, I'm particularly curious about how sensitive the performance is to the RFF approximation quality. Would it be possible to see the actual results?
> > So, I’ll reserve my judgment for the moment.

---

> > > ### Author Response · Authors · 2026-04-03
> > >
> > > Dear reviewer,
> > >
> > > Thank you for clarifying the ethics flag issue.
> > >
> > > > I'm particularly curious about how sensitive the performance is to the RFF approximation quality. Would it be possible to see the actual results?
> > >
> > > We may be misunderstanding your question, so please let us know if we do not correctly interpret it. One interpretation is that you are looking for an ablation over the number of RFF features for our proposed method, the Alpha-PFN.
> > >
> > > **Computationally not feasible**
> > >
> > > If an ablation is intended, it would require regenerating the pretraining data (re-optimizing ~90M GP sample paths), since changing the RFF approximation alters paths and their optima’s. This is unfortunately not feasible within the rebuttal timeline because of the compute time required. In our final model, we have settled on 500 RFFs.
> > >
> > > **Justified default**
> > >
> > > As mentioned in our rebuttal, we did not systematically vary this parameter. However, RFF approximation quality, for the variant we use, is well understood theoretically [3], and our choice of 500 features is already relatively large. See also https://archives.argmin.net/2017/12/05/kitchen-sinks/ who mention: “But in all of our experiments, we were getting great results with a only few hundred random features.” suggesting 500 is sufficient.
> > >
> > > **Comparison with the GP**
> > >
> > > Finally, it should be remarked that for the MES, JES and PES approximations, a number of 1024 RFF’s is used in the approximation (as per default in Botorch). Our method instead is using 500 RFF’s. This clearly favors the original MES, JES and PES approximations instead of our proposed Alpha-PFN.
> > >
> > > **Please clarify**
> > >
> > > Could you please clarify: why would you like us to vary the number of RFFs? Would you be interested to see results with more or with less RFFs (or both)?
> > >
> > > Note that, we will not be able to see your reply if you comment directly on this response. Could you be so kind as to let us know by editing your original review? We would be curious to know what you would like to see for the ablation, so we can consider it for future work.
> > >
> > > Many thanks!

---

### Official Review · Reviewer_bCms · 2026-03-13

**Soundness:** 4
**Presentation:** 4
**Significance:** 3
**Originality:** 2
**Overall Recommendation:** 5
**Confidence:** 4

**Summary:**

This work shows that Prior Fitted Networks (PFNs) can be trained to closely mimic the behaviour of the standard suite of Entropy Search Bayesian Optimisation methods (Entropy Search, Joint Entropy Search and Max-Value Entropy Search). The authors describe a family of loss functions and in-context learning assumptions that allow them to amortize the otherwise highly compute intensive decision making of the standard entropy search algorithms. They provide theoretical justification for their method, and very comprehensive empirical study of their trained networks against SotA baseline entropy search implementations.

**Compliance With Llm Reviewing Policy:**

Affirmed.

**Key Questions For Authors:**

Did you perform any scaling-law style experiments when training your alpha-PFNs? In particular, do you have a sense for how important it is for practitioners to come close to 90M GP realisations when training their models? How much does performance drop using 10x fewer samples?

**Limitations:**

The authors acknowledge that their method trades online compute savings for increased offline compute burden, and that they only seek to compare against *ES methods and so they are not attempting to make a SotA claim w.r.t. regret. Overall their limitations are fair and reasonable.

**Strengths And Weaknesses:**

While the overall focus of this paper is not entirely novel, (as the authors point out, several papers have focused on amortizing expensive information theoretic policies into a cheaper feedforward process), this paper is a valuable contribution to the deep learning and Bayesian optimisation community for its clarity, focus and scale of experimentation.

In particular, the authors' primary results showing that alpha-PFNs can very closely mimic the performance of popular entropy search methods at substantially reduced test-time compute burdens is impressive. The strategy to approximate the distribution of on-policy information sets is a useful trick to improve efficiency without having to embrace a full on-policy RL style training setup.

Their overview of the related literature is, for the most part, accurate and representative of the relevant literature (I would encourage the authors to additionally cite [1] which is relevant though takes a different focus). Their experimental section is particularly comprehensive and includes the most relevant baselines and their appropriate implementations. I found this paper to be very well thought out, very well written, and a refreshingly comprehensive treatment of the topic.


[1] Efficient Bayesian Experiment Design with Equivariant Networks, Igoe et al

---

> ### Author Rebuttal · Authors · 2026-03-31
>
> Thank you for your positive review.
>
> **Did you perform any scaling law style experiments**
>
> We did not run a systematic scaling law study. We also did not run a controlled model-size ablation. What we did observe during development is that scaling the amount of training data improves performance, especially for MES, when going from roughly 1M to 10M to 90M GP samples. This aligns with previous PFN work [1], which compares MCMC against varying PFN training setups. [1] concludes that the amount of training is more important than the model size. We will clarify this in the revision.
>
> **Please cite Efficient Bayesian Experiment Design with Equivariant Networks**
>
> We will cite Efficient Bayesian Experiment Design with Equivariant Networks by Igoe et al. (2025). Besides the architectural difference, Igoe et al. consider learning non-myopic policies using reinforcement learning, while our work focuses on amortizing existing information theoretic acquisition functions in a supervised setting.
>
> **Offline cost versus online savings should be clearer**
>
> Thanks for your feedback. The detailed training costs are detailed in Appendix F. For the revised version, we will move it to the main text and add a discussion when the approach is expected to be practical compared to GP.
>
>
> [1] Steven Adriaensen, Herilalaina Rakotoarison, Samuel Müller,  Frank Hutter. Efficient Bayesian Learning Curve Extrapolation using Prior-Data Fitted Networks. NeurIPS 2025.
>
> ________
>
> We hope you find these responses satisfactory. If you have additional questions or concerns, please do not hesitate to raise them.

---

> > ### Author Rebuttal · Reviewer_bCms · 2026-04-06
> >
> > I appreciate the response from the authors. I will keep my original score.

---

> > > ### Author Response · Authors · 2026-04-07
> > >
> > > Again, thanks for your time and your feedback!

---

### Decision · Program_Chairs · 2026-04-30

**Decision:**

Accept (regular)

**Comment:**

The paper proposes alpha-PFN which is a two-stage amortization approach that approximates entropy search acquisition functions (in a single forward pass using Prior-data Fitted Networks. All reviewers agreed that substantial speedup of entropy search would enable more use of these acquisition functions. Reviewer Ygzd's concerns about prior mismatch and offline training costs apply to amortized inference broadly rather than this specific paper. Overall, the key idea is conceptually new and interesting and the evaluation is comprehensive.  **Therefore,I recommend accepting the paper.** I encourage the authors to incorporate all comments from the reviewers in the final version.